# SMC complex unidirectionally translocates DNA by coupling segment capture with an asymmetric kleisin path

**Masataka Yamauchi, Giovanni Bruno Brandani, Tsuyoshi Terakawa, Shoji Takada\***

Department of Biophysics, Graduate School of Science, Kyoto University, Kitashirakawa Oiwakecho, Kyoto, Japan

## eLife Assessment

This **important** study presents a well-constructed multiscale simulation framework to investigate ATP-driven DNA translocation by prokaryotic SMC complexes, supporting a segment-capture mechanism. The strength of evidence is **convincing**, highlighting the necessity of a precise balance between electrostatic interactions and hydrogen bonding, as well as the critical role of kleisin asymmetry in ensuring unidirectional movement.

**\*For correspondence:**
takada@biophys.kyoto-u.ac.jp

**Abstract** SMC (structural maintenance of chromosomes) protein complexes are ring-shaped molecular motors essential for genome folding. Despite recent progress, the detailed molecular mechanism of DNA translocation in concert with the ATP-driven conformational changes of the complex remains to be clarified. In this study, we elucidated the mechanisms of SMC action on DNA using all-atom and coarse-grained molecular dynamics simulations. We first created a near-atomic full-length model of a prokaryotic SMC–kleisin complex based on experimental structures and implemented ATP-dependent conformational changes using a structure-based coarse-grained model. We further incorporated key protein–DNA hydrogen-bond interactions derived from fully atomistic simulations. Extensive simulations of the SMC complex with 800 base pairs of duplex DNA over the ATP cycle observed unidirectional DNA translocation by the SMC complex. The process exhibited a step size of ~200 base pairs, wherein the SMC complex captured a DNA segment of about the same size within the SMC ring in the engaged state, followed by its pumping into the kleisin ring as ATP was hydrolyzed. Analysis of trajectories identified the asymmetric path of the kleisin as a critical factor for the observed unidirectionality.

## Introduction

The structural maintenance of chromosomes (SMC) complexes are protein complexes essential to organize chromosome structures and are highly conserved across the three domains of life (*Davidson and Peters, 2021*; *Higashi and Uhlmann, 2022*; *Hirano, 2006*; *Kim et al., 2023*; *Yatskevich et al., 2019*). Eukaryotic SMCs include condensin, which plays a significant role in mitotic chromosome architecture; cohesin, known for its dual roles in sister chromatid cohesion and chromosome folding during interphase; and SMC5/6, implicated in DNA damage repair, chromosome segregation, and replication processes (*Aragón, 2018*). Bacteria and archaea also possess SMC complexes whose architecture resembles their eukaryotic counterparts (*Nolivos and Sherratt, 2014*).

All SMC complexes have a common ring-shaped architecture of two structured SMC subunits and a kleisin subunit, which is mainly disordered (*Hirano, 2006*; *Yatskevich et al., 2019*). Each SMC subunit has an ATPase head domain and a hinge domain, which are connected by a ~50 nm coiled-coil arm.

Two SMC chains dimerize through their hinge domains, while the kleisin bridges the two ATPase head moieties by binding through its N- and C-termini, thus forming a closed ring. Eukaryotic SMC chains form heterodimers, SMC1 and SMC3 for cohesin, SMC2 and SMC4 for condensin, and SMC5 and SMC6 for the SMC5/6 complex, while the prokaryotic ones form homodimers. Other non-SMC subunits, KITE (kleisin-interacting tandem winged-helix element) and HAWK (HEAT repeat proteins associated with kleisins), can be further recruited onto the kleisin subunit.

SMC proteins are also characterized by conformational changes coupled with the ATP hydrolysis cycle (*Davidson and Peters, 2021*; *Higashi and Uhlmann, 2022*; *Hirano, 2006*; *Kim et al., 2023*; *Yatskevich et al., 2019*). In the ATP-bound state (designated as the engaged state hereafter), two ATPase heads are engaged to form a dimer to catalyze ATP hydrolysis. The two coiled-coil arms separately emanate from the engaged heads and converge at the hinge domains, forming an O-shaped ring structure. In the ATP-bound state, a second ring is formed by the kleisin as it links together the head domains of each SMC through its intrinsically disordered region. After ATP hydrolysis, the two head domains dissociate, with the coiled-coil arms forming a V-shaped structure (designated as the V-shape state). In the nucleotide-free state, the two ATPase heads associate again, but differently from the engaged state, in which two coiled-coil arms become aligned in parallel, forming an I-shaped structure (designated as the disengaged state). Notably, throughout the hydrolysis cycle, the kleisin subunit keeps bridging two ATPase head moieties, maintaining the overall loop topology.

These ATP-dependent conformational changes have been suggested to realize the SMC motor activities responsible for higher-order chromosome organization (*Alipour and Marko, 2012*; *Fudenberg et al., 2016*; *Goloborodko et al., 2016b*; *Nasmyth, 2001*). Direct evidence that SMC complexes act as ATP-dependent molecular motors was first obtained from single-molecule imaging studies on budding yeast condensin (*Terakawa et al., 2017*), revealing its unidirectional translocation along DNA. Single-molecule imaging studies later demonstrated DNA-loop extrusion by condensin (*Ganji et al., 2018*), cohesin (*Davidson et al., 2019*; *Kim et al., 2019*), and SMC5/6 complexes (*Pradhan et al., 2023*). Time-resolved Hi-C and ChIP-seq experiments also provided evidence of ATP-dependent translocation activity of bacterial SMC complexes (*Wang et al., 2018*; *Wang et al., 2017*).

While the ATP-dependent conformational changes and biochemical activities of SMC complexes have been characterized, the link between the two remains elusive, with several models proposed to delineate the molecular mechanism of DNA translocation and loop extrusion (*Davidson and Peters, 2021*; *Higashi and Uhlmann, 2022*; *Kim et al., 2023*). The 'DNA-segment capture' model (*Diebold-Durand et al., 2017*; *Marko et al., 2019*; *Nomidis et al., 2022*) was originally inspired by the structural features of prokaryotic SMC complexes: the I-shaped conformation in the apo state and open-ring conformation in the ATP-bound state. In this model, DNA-loop extrusion is achieved by translocating along DNA while anchoring it at an additional binding site, presumably through a safety belt (*Kschonsak et al., 2017*). This DNA-segment capture model has been extended to eukaryotic SMC complexes. DNA entrapment in the kleisin and two SMC sub-compartments, consistent with the segment capture mode, was experimentally demonstrated for the SMC5/6 holo-complex (*Taschner and Gruber, 2023*). In contrast, a revised 'hold-and-feed' model was proposed to explain observations on condensin (*Shaltiel et al., 2022*). For the DNA-loop extrusion mechanism of condensin and cohesin, the 'scrunching' (*Ryu et al., 2022*; *Ryu et al., 2020*), 'swing-and-clamp' (*Bauer et al., 2021*), and 'Brownian ratchet' (*Higashi et al., 2021*) models, which emphasize the folding of the coiled-coil arms at the elbow joints, have also been proposed. However, the detailed mechanism by which the ATP-dependent conformational changes of the SMC complexes are coupled with DNA translocation has not yet been thoroughly tested.

In this study, we aimed to understand the molecular mechanism of DNA translocation by an SMC complex using molecular dynamics (MD) simulations. In contrast to previous work (*Brandão et al., 2021*; *Nomidis et al., 2022*; *Takaki et al., 2021*), we employed a bottom-up multiscale approach where DNA dynamics arise from the fundamental physical interactions of the system. Based on experimental structural knowledge, we built an energy landscape of SMC complex for each of the bound nucleotide state and simulated ATP-dependent conformational changes by switching energy landscapes in the order of the ATP hydrolysis cycle in a similar manner to previous works (*Astumian, 1997*; *Brandani and Takada, 2018*; *Hyeon et al., 2006*; *Hyeon and Onuchic, 2007*; *Koga and Takada, 2006*; *Nomidis et al., 2022*; *Pu and Karplus, 2008*; *Tully, 1990*). In this approach, DNA dynamics is purely derived from physical interaction with the SMC complexes in an unbiased manner without

assuming any specific DNA translocation model a priori. As a relatively simple model system broadly representative of SMC complexes, we focused on prokaryotic SMC, which is made of the SMC homodimer and the kleisin subunit, ScpA.

We first performed fully atomistic MD simulations with explicit water solvent to uncover fundamental hydrogen-bond interactions between SMC and duplex DNA. Next, we conducted coarse-grained MD simulations using a rigorous physics-based modeling approach at near-atomic resolution, realizing the cyclic conformational changes occurring in the full-length SMC–ScpA complex throughout ATP binding and hydrolysis. We then characterized the key DNA-binding sites in the SMC complex through simulations in the presence of DNA, incorporating long-ranged Coulombic interactions based on the Debye–Hückel approximation and short-ranged hydrogen bonds based on the atomistic MD simulations. Finally, we performed extensive simulations where the SMC complex undergoes cyclic conformational changes as an 800 base pairs (bp) long DNA molecule is loaded inside the SMC complex, successfully observing unidirectional DNA translocation. Analysis of our MD trajectories reveals that translocation proceeds via DNA-segment capture within the SMC ring and its subsequent pumping to the kleisin ring, with the asymmetric path of the kleisin being responsible for the unidirectionality. We also analyzed many non-productive trajectories to discuss the inherent difficulty of efficient DNA translocation.

## Results

### Interactions between prokaryotic SMC ATPase heads and DNA

Recent cryo-EM structures have uncovered highly conserved DNA-clamping states among SMC complexes (*Bürmann et al., 2021*; *Collier et al., 2020*; *Lee et al., 2022*; *Shi et al., 2020*; *Yu et al., 2022*), suggesting that DNA binding to the upper surface of the ATPase heads is crucial for SMC functionality. As prokaryotic SMC ATPase–DNA complexes have not been experimentally resolved at high resolution, we first studied this system using all-atom MD simulations. The simulations allowed us to identify the hydrogen bonds formed between DNA and key basic residues on SMC and to incorporate these interactions (*Niina et al., 2017*) into coarse-grained simulations for the following stages of our investigation. Based on the available biochemical evidence (*Vazquez Nunez et al., 2019*), we prepared an initial structure by placing a 47-bp duplex DNA fragment on the upper surface of engaged *Pyrococcus furiosus* (*Pf*) SMC ATPase heads (PDB code: 1xex) (*Lammens et al., 2004*) and conducted five independent 1-µs-long simulations (refer to Methods for more details). Starting from its initially straight conformation, the DNA adopted moderately bent conformations as the bending increased DNA interactions with the head domains (*Figure 1A*, upper panel, and *Video 1* for a representative trajectory). DNA binding to the top head surface remained stable for the entire simulation in four out of five trajectories. In one case, one DNA end detached from the heads in the middle of the simulation, losing most interactions on that side, so this trajectory was excluded from the subsequent interaction analysis.

During our trajectories, the SMC ATPase heads gradually formed hydrogen bonds with the DNA backbone, reaching an equilibrium with 13 ± 3 bonds on average after ~500 ns (*Figure 1A*, lower panel, and *Figure 1—figure supplement 1*). In subsequent analyses, we focused on the data after 500 ns. *Figure 1B* shows the probability of hydrogen-bond formation, highlighting many critical residues on the top head surface stabilizing the interaction with DNA (*Figure 1C*, right panel). In particular, basic residues Arg120, Arg123, Arg111, Arg62, and Lys56 frequently formed hydrogen bonds (*Figure 1B, C*, left panel). The critical role of these residues is consistent with previous experiments by Vazquez Nunez et al. demonstrating that the triple alanine mutations of Lys60, Arg120, and Lys122 in *Bacillus subtilis* (*Bs*) SMC (which correspond to Arg62, Arg120, and Arg123 in *Pf*SMC) result in impaired DNA-binding ability and a severe growth phenotype (*Vazquez Nunez et al., 2019*). Also, these amino acid residues are consistently detected across different force field sets, as shown in *Figure 1—figure supplement 2*, indicating the robustness of the all-atom force field (see Appendix 1 for details). The identification of these important basic residues and their consistency with experimental findings supports the reliability of all-atom simulations. We will later use these findings to optimize the representation of protein-DNA interactions in our residue-resolution model of ATP-dependent DNA translocation.

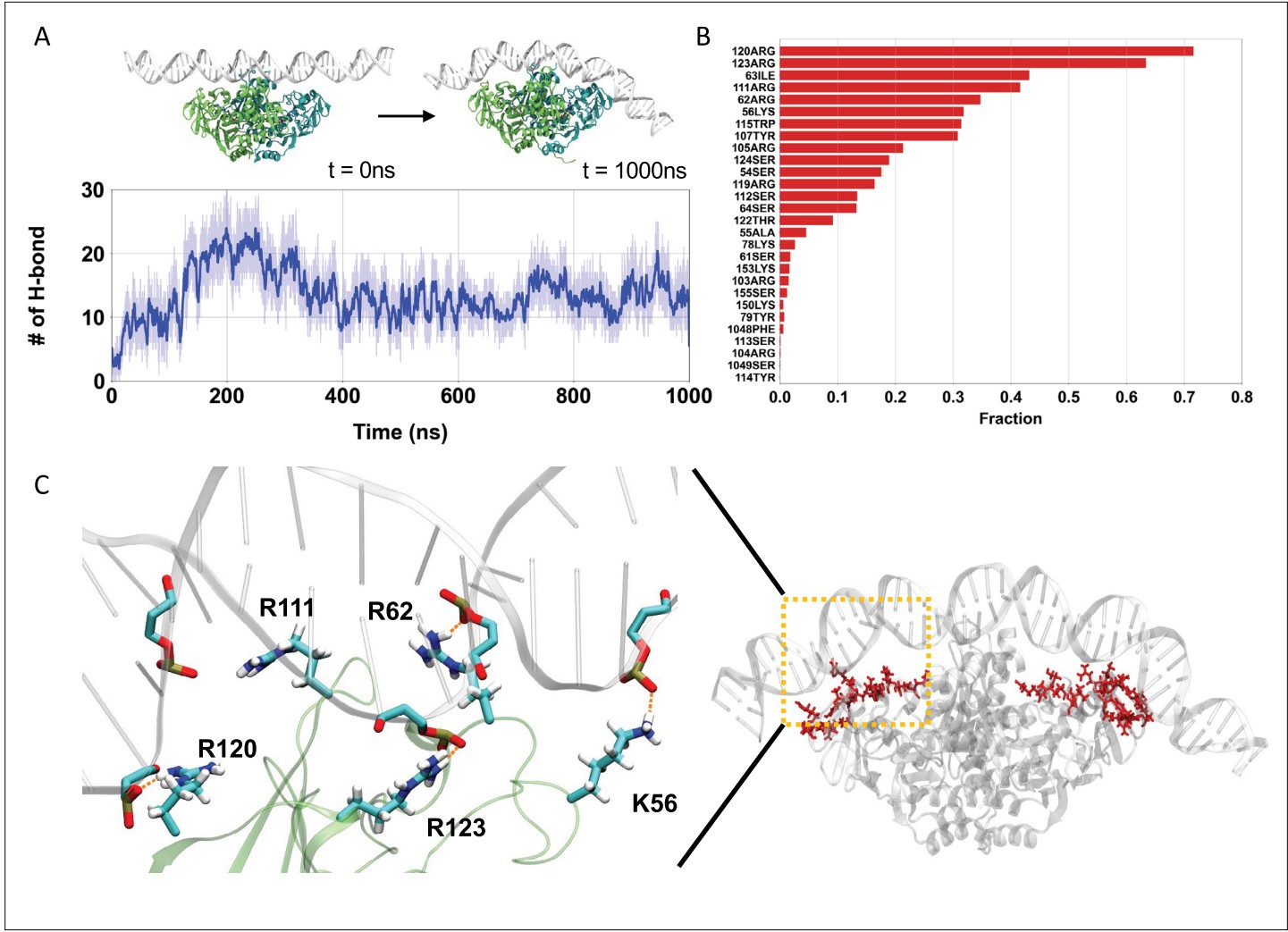

**Figure 1.** All-atom molecular dynamics (MD) simulations of engaged *Pf*SMC ATPase heads and DNA. (**A**) A representative trajectory of all-atom MD. (Upper) Left and right depict the initial and final snapshots, respectively. (Lower) Time series of the number of hydrogen bonds between ATPase heads and DNA. (**B**) The fraction of hydrogen-bond formation between amino acid residues in the ATPase heads and DNA. (**C**) A representative snapshot of the hydrogen-bond forming residues. (Left) Hydrogen-bond network between the ATPase heads and DNA for representative basic residues. (Right) The hydrogen-bond forming residues are positioned on the top surface of ATPase heads.

The online version of this article includes the following figure supplement(s) for figure 1:

**Figure supplement 1.** Time series of the number of hydrogen bonds between *Pf*SMC ATPase and DNA derived from all-atom molecular dynamics (MD) simulations.

**Figure supplement 2.** Force field dependence of tendency of hydrogen-bond formation between SMC ATPase heads and DNA.

**Figure supplement 3.** Probability distribution of the coarse-grained hydrogen-bond parameters, $r$, $\theta$, and $\varphi$ for frequently hydrogen-bond forming residues.

## Modeling the full-length SMC–ScpA complex in the disengaged form

Next, we turn to the problem of modeling the full-length prokaryotic SMC complex (*Krepel et al., 2020*; *Krepel et al., 2018*). A previous biochemical study showed that in several archaea, such as *Pyrococcus yayanosii* (*Py*), a binary complex (SMC–ScpA) is more likely to form than a tripartite complex (SMC–ScpAB) due to the absence of the ScpB (KITE)-binding segment in the ScpA subunit (*Jeon et al., 2020*). Therefore, our current study focuses on the SMC–ScpA binary complex as a minimal configuration to investigate the translocation of SMC complexes.

To prepare the initial configuration for our MD simulations, we built a full-length model of SMC–ScpA in the disengaged (I-shaped) form. We considered the full-length *Py*SMC homodimer model

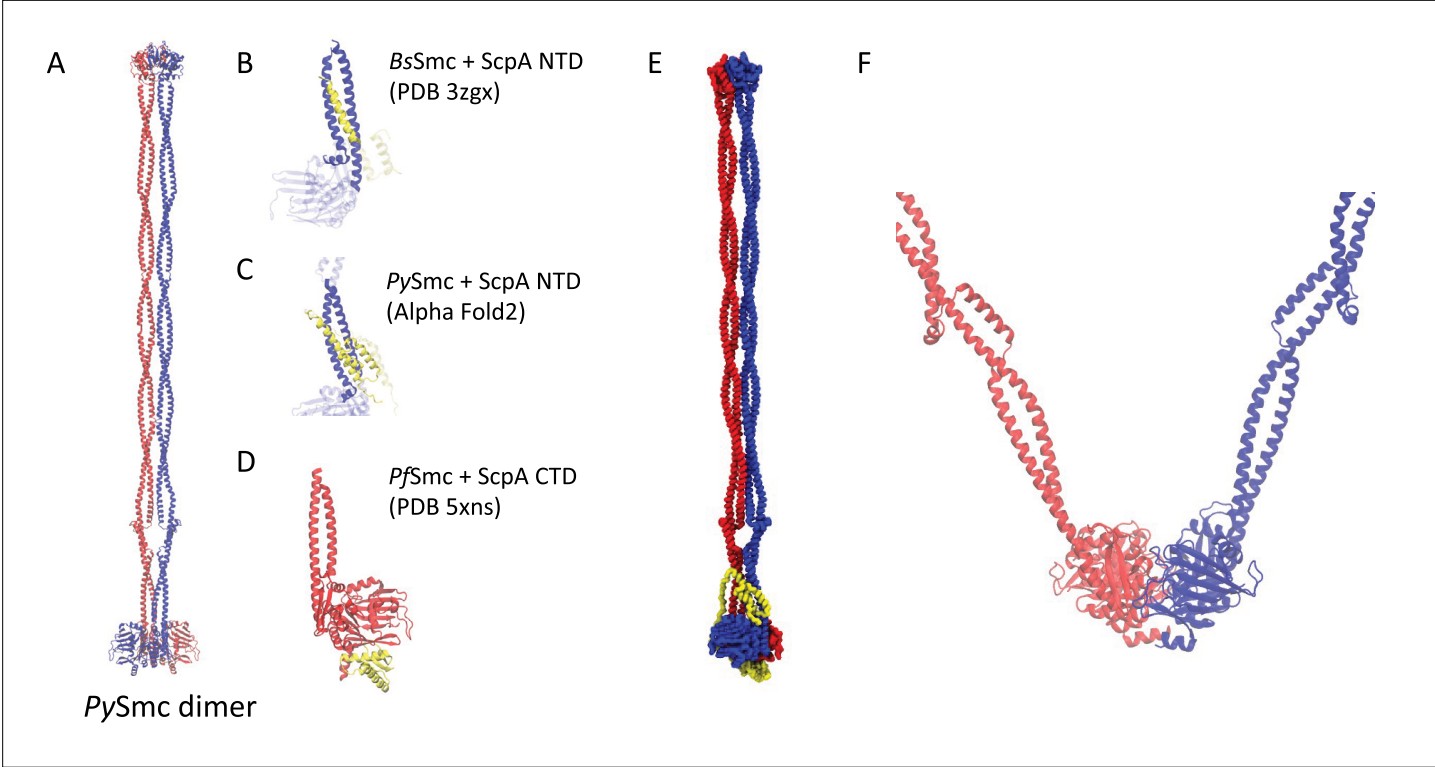

**Figure 2.** Modeling of a full-length *Py*SMC–ScpA complex. (**A–D**) Template structures for homology modeling of a full-length *Py*SMC–ScpA complex. The regions ignored in the homology modeling are indicated by transparent. (**E**) A coarse-grained full-length *Py*SMC–ScpA complex model where one amino acid is represented by one bead. (**F**) A homology model of the engaged ATPase heads based on a crystal structure (PDB code: 1xex). The coiled-coil arms connected to the ATPase heads are modeled based on the I-shaped SMC dimer model in panel A.

(*Figure 2A*), which was previously constructed based on protein cross-linking experiments and crystal structures of individual domains (*Diebold-Durand et al., 2017*), as a reference structure. We added the ScpA subunit to this model using Modeller 10.1 (*Webb and Sali, 2016*). Here, crystal structures (PDB codes: 3zgx and 5xns) and an AlphaFold2 predicted complex structure were used as templates, which comprises the ScpA N-terminal domain (NTD) and C-terminal domain (CTD) bound to the SMC ATPase head domain (*Figure 2B–D*). Details are provided in the Methods section. The resulting full-length model of SMC–ScpA in the disengaged form is shown in *Figure 2E*. Our modeling generated the expected ring topology via the binding of the ScpA NTD to one SMC subunit and the ScpA CTD to the other. We note that while SMC forms a homodimer, the binding to ScpA breaks the original symmetry of the structure. For convenience, hereafter, we distinguish the two subunits by color; the blue subunit interacts with NTD, while the red subunit interacts with CTD of ScpA.

## MD simulations of SMC–ScpA ATP-dependent conformational changes

During ATP hydrolysis, the SMC complex undergoes cyclic conformational changes as it transitions among three nucleotide-binding states: Apo (disengaged), ATP-bound (engaged), and ADP-bound (V-shape) states (*Kamada et al., 2017*; *Yatskevich et al., 2019*). Due to the complex molecular size (over 2000 amino acid residues and ~50-nm-long coiled-coils) and large-scale motions involved, all-atom MD simulations are computationally too costly to observe these conformational changes. To overcome this issue, we employed the structure-based AICG2+ coarse-grained model at residue resolution (*Li et al., 2014*), where each amino acid in proteins is represented by one bead located on the Cα atom. The combination of this protein model and the 3SPN.2C model for DNA has been successfully applied to investigate many protein–DNA complexes (*Brandani et al., 2022*; *Brandani et al., 2018*; *Nagae et al., 2023*; *Nagae et al., 2021*; *Tan and Takada, 2020*).

Following the energy landscape theory of proteins, we can effectively simulate the ATP-dependent conformational changes in the SMC complex by switching the energy landscape centered to the

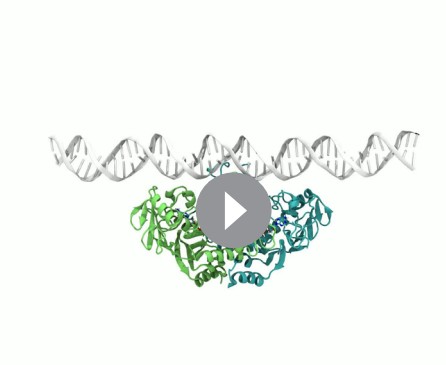

**Video 1.** The representative trajectory of all-atom molecular dynamics (MD) simulations for engaged *Pf*SMC ATPase heads and DNA.

https://elifesciences.org/articles/106752/figures#video1

reference structures in the model (*Brandani and Takada, 2018*; *Hyeon et al., 2006*; *Hyeon and Onuchic, 2007*; *Kamada et al., 2017*; *Koga and Takada, 2006*). In the energy landscape theory of proteins, ATP-driven biomolecular machines explore distinct landscapes depending on bound nucleotides, and the ATP-driven motions can be simulated by making transitions among different energy surfaces based on the well-known order of chemical events. For each nucleotide state, an energy landscape is concisely represented by a structure-based potential guided by its stable structure information. We note that the predictions of such switching energy landscape simulations have often been successfully validated against independent experimental data. For example, single-molecule FRET experiments on chromatin remodelers (*Sabantsev et al., 2019*) confirmed the mechanism suggested by previous potential-switching simulations (*Brandani and Takada, 2018*).

The choice of reference structure in each ATP state is based on experimental observations established in past structural studies on SMC complexes. For the disengaged state, the reference structure of the full-length SMC–ScpA is the one we have built above. For the engaged state, we changed the reference structure of the ATPase heads to that of the corresponding crystal structure of *Pf*SMC (PDB code: 1xex, *Figure 2F*). For the V-shape structure, we used the same reference structure as the disengaged one, except we turned off the inter-subunit interactions between the ATPase head domains, as these are not expected to be stably interacting in this state (*Hirano, 2001*; *Hirano and Hirano, 2006*; *Kamada et al., 2017*; *Melby et al., 1998*). During the MD simulations, the potential energy function was switched in the following order: disengaged, engaged, V-shape, and disengaged, with each state lasting for $5 \times 10^7$ MD steps.

*Figure 3* and *Video 2* show a representative trajectory illustrating the ATP-dependent conformational changes of the SMC–ScpA complex. In the initial disengaged state, the entire SMC complex adopted the I-shaped structure where two ATPase heads were in contact, and the coiled-coil arms were aligned in parallel (*Figure 3A*). Upon transition to the engaged state, the two ATPase heads were quickly rearranged to form the new inter-subunit contacts. Specifically, this rearrangement involves one ATPase head sliding by approximately 10 Å and rotating by 85° relative to the other, allowing it to associate through a different interface (*Diebold-Durand et al., 2017*; *Hyeon and Thirumalai, 2005*). The fractions of formed contacts, *Q*-scores, that exist at the disengaged (engaged) states quickly decreased (increased) (*Figure 3A*, top two plots). The head dimer rearrangement induced the opening of the adjacent coiled-coil arms, which further propagated to open the hinge domain, as monitored by the hinge angle (*Figure 3A*, bottom plot). The resulting configuration was an O-shaped structure. The two ATPase heads dissociated in the subsequent V-shape state, while the dimerized hinge domain maintained an open hinge conformation (*Figure 3A*, bottom two plots). Finally, the SMC–ScpA complex returned to the disengaged conformation upon re-establishing the inter-subunit interactions between the ATPase head domains. *Figure 3B* shows the average distances between intermolecular amino acid residues to characterize typical conformations in each nucleotide state. This is computed from the distance between a Cα atom of the *i*th residue for the first SMC protein and a Cα atom of *i*th residue for the second SMC protein. The results clearly represent that the disengaged state has a rod-shaped structure, the engaged state has an open-ring structure, and the V-shape state has dissociated ATPase heads and a dimerized hinge domain. These structural features are consistent with experimental observations by electron micrographs (*Kamada et al., 2017*; *Melby et al., 1998*).

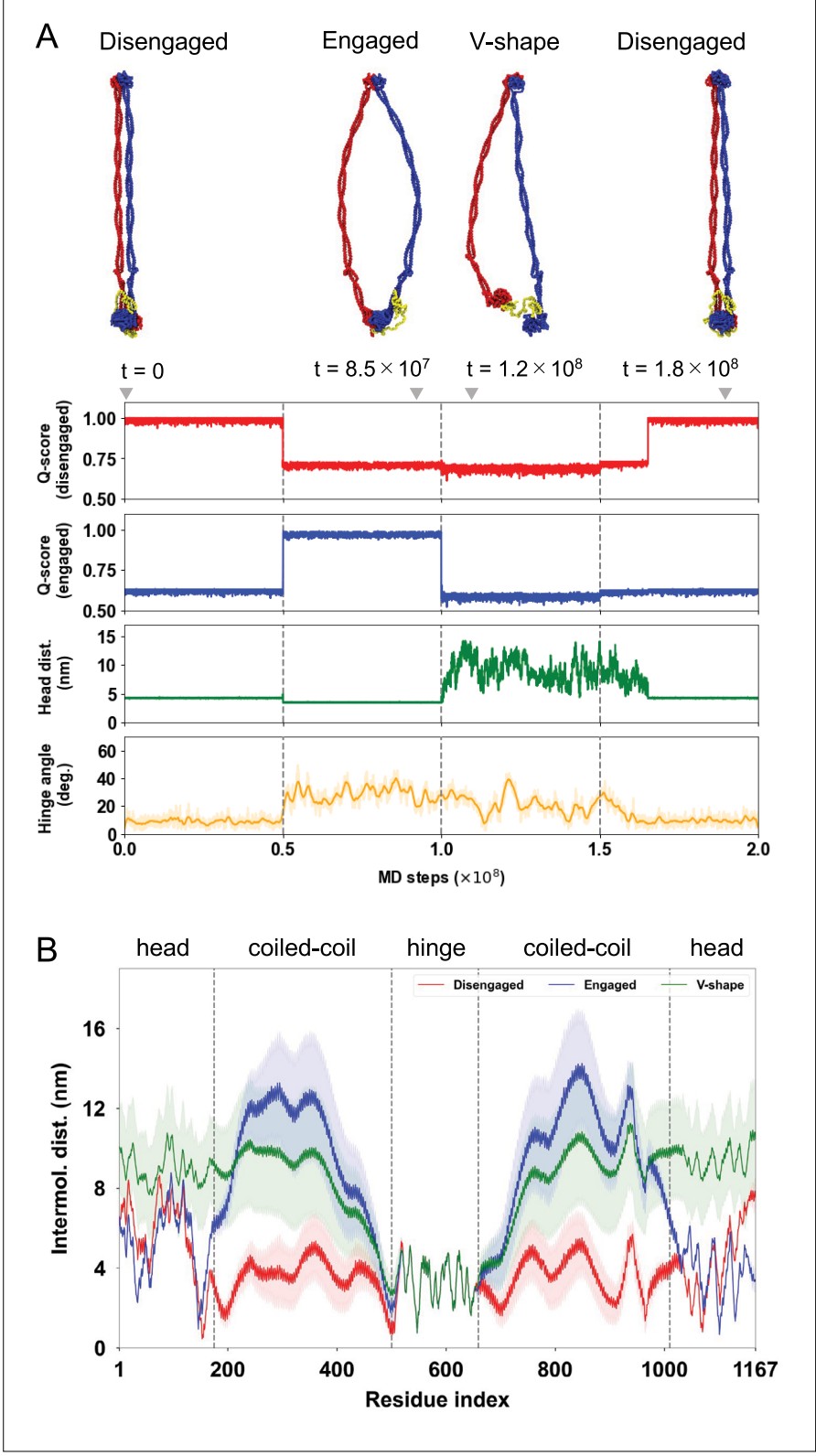

**Figure 3.** Molecular dynamics (MD) simulations for ATP-dependent conformational changes of SMC–ScpA complex. (**A**) A representative trajectory of ATP-dependent conformational changes in the SMC complex by switching the reference structures in the AICG2+ model. (Top) Typical snapshots of SMC–ScpA complex for the disengaged ($t = 0$ step), engaged ($t = 8.5 \times 10^7$ steps), V-shape ($t = 1.2 \times 10^8$ steps), and disengaged ($t =$

*Figure 3 continued on next page*

*Figure 3 continued*

$1.8 \times 10^8$ steps) states. (Bottom) Times series of Q-scores for the disengaged and engaged structure, head-to-head distance, and hinge angle. The head-to-head distance is defined as the center of distance between two ATPase head domains. The hinge angle is calculated for three selected points; the center of mass of the hinge dimerization domain defines one point. The other two points are defined by the center of mass of the sequential 10 amino acid residues in the coiled-coil middle region for each chain. (**B**) Average distances between intermolecular amino acid residues with the same index number for the disengaged (red), engaged (blue), and V-shape (green) structures.

## DNA-binding sites in the SMC–ScpA complex

The DNA-binding sites on the SMC complex are expected to play critical roles in DNA translocation and loop extrusion, but these were not comprehensively investigated for prokaryotic SMC. Here, we characterize DNA binding in the full-length SMC–ScpA complex using residue-resolution MD simulations, a similar approach previously employed for yeast condensin (*Koide et al., 2021*).

To set the initial structures for the DNA-binding simulations, we distributed five 40 bp double-strand (ds) DNAs with identical random sequences as used in prior fluorescence anisotropy measurements (*Vazquez Nunez et al., 2019*) around the engaged SMC–ScpA complex. The DNA was modeled using the 3SPN2.C coarse-grained model (*Freeman et al., 2014*), where each nucleotide was represented by three beads, each located at the base, sugar, and phosphate groups. Protein and DNA interact through excluded volume, Debye–Hückel electrostatics, and an effective model of hydrogen-bond interactions (*Niina et al., 2017*) with parameters obtained from the previous all-atom MD simulations (more details are provided in the Methods section). We conducted 100 simulation runs of $5 \times 10^7$ steps, each starting from a different initial distribution of DNA molecules, all at 150 mM monovalent ion concentration (simulations at 300 mM did not change major binding sites although the bindings were weakened as expected).

*Figure 4A, B* shows the DNA contact probabilities as a function of SMC residue. The analysis revealed three major DNA-binding sites on the SMC heads, coiled-coil arms, and hinge domain. The ScpA subunit also possesses a DNA-binding patch in its disordered region due to the consecutive arrangement of positively charged lysine residues (at positions 126–130). *Figure 4C* illustrates a representative snapshot of DNAs that bound to the SMC–ScpA complex; here, DNA was observed to be sandwiched between the coiled-coil arms on the inner side of the hinge, consistent with previous biochemical experiments (*Griese and Hopfner, 2011*; *Hassler et al., 2018*; *Hirano and Hirano, 2006*; *Vazquez Nunez et al., 2019*). The DNA also binds to the top surface of ATPase heads, which aligns with our all-atom simulations and previous biochemical experiments (*Vazquez Nunez et al., 2019*). *Figure 4D–H* and *Video 3* illustrates a typical DNA-binding event on the top surface of ATPase heads: DNA initially bound to the side surface of the heads (*Figure 4E, F*) and subsequently migrated to the top surface (*Figure 4G, H*) driven by the formation of hydrogen bonds (*Figure 4D*, bottom plot). It is worth noting that disabling hydrogen-bond interactions resulted in DNA remaining attached to the side surface of the ATPase heads (*Figure 4—figure supplement 1*). This highlights the importance of hydrogen bonds in accurate modeling of the DNA-head binding, as long-range electrostatic interactions alone were insufficient.

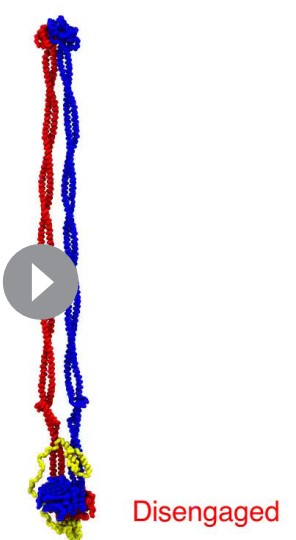

**Disengaged**

**Video 2.** The representative trajectory of ATP-dependent conformational changes of SMC–ScpA complex.

https://elifesciences.org/articles/106752/figures#video2

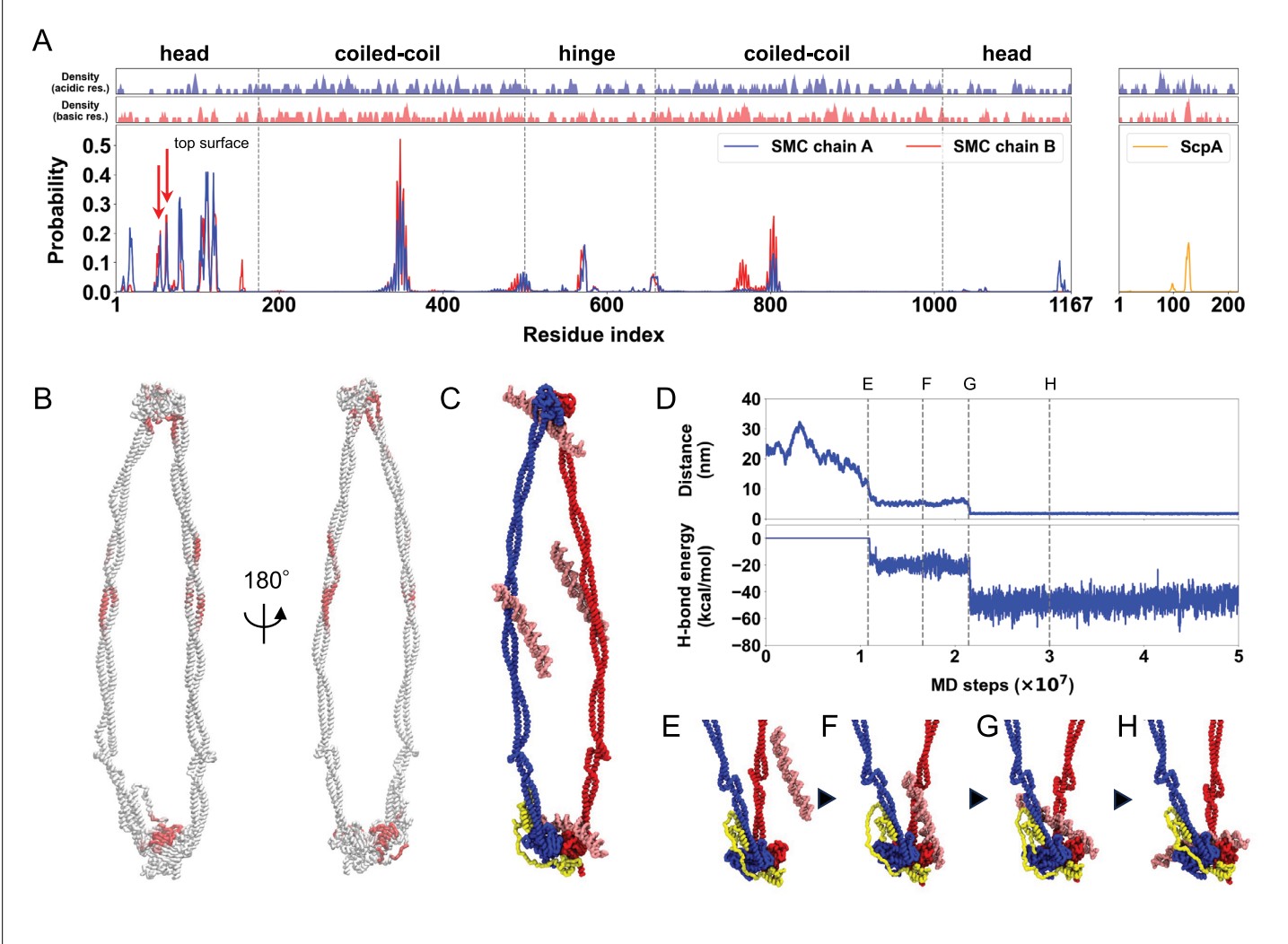

**Figure 4.** Identification of DNA-binding sites in the SMC–ScpA complex. (**A**) Top two panels plot the local average of charges defined as the moving average with a window size of five residues. Bottom panel plots the contact probability between DNA and SMC–ScpA complex. (**B**) DNA contact probability mapped on the SMC–ScpA structure. (**C**) A typical snapshot of DNA binding to the SMC–ScpA complex. (**D**) Upper panel plots timeseries of the distance between center of mass of the ATPase heads and DNA. Lower panel plots timeseries of the hydrogen-bond energy. (**E–H**) Representative snapshots during a DNA-binding event to the top of the SMC ATPase heads.

The online version of this article includes the following figure supplement(s) for figure 4:

**Figure supplement 1.** DNA-binding sites in the SMC–ScpA complex where hydrogen-bond interactions on the ATPase heads are not incorporated.

## DNA translocation via DNA-segment capture

We employed the established full-length SMC–ScpA model to address how the SMC complex couples ATP-dependent conformational changes with DNA binding to realize unidirectional SMC translocation. To this end, we conducted the same cyclic conformational change simulations of SMC–ScpA with a long (800 bp) dsDNA topologically threaded into the ring complex. We started our simulations in the disengaged state with DNA placed into the kleisin ring (*Figure 5—figure supplement 1*; *Vazquez Nunez et al., 2019*), switched to the engaged, then V-shape, and again disengaged state. In the production simulations, we performed 50 individual simulations for the first disengaged and subsequent engaged state. Due to its stochastic nature, for subsequent V-shape and disengaged states, we repeated multiple simulations for each selected final structure with different realizations of fluctuating forces; consequently, the number of trajectories increased to 190 and 770 for the V-shape and the last disengaged state, respectively, allowing us to explore the key processes. Here, the monovalent

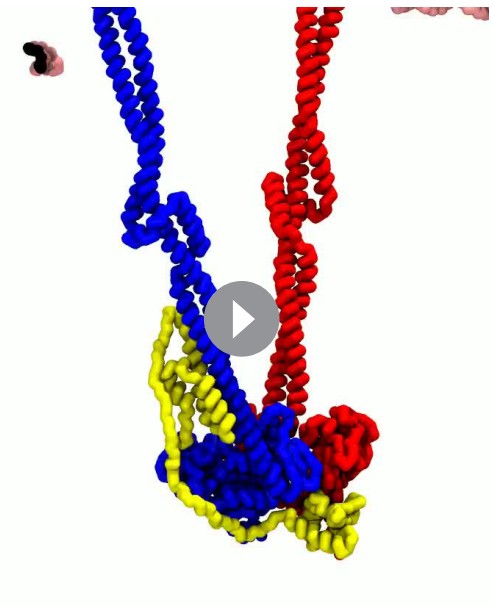

**Video 3.** The representative trajectory of DNA-binding events to the top of ATPase heads.

https://elifesciences.org/articles/106752/figures#video3

ion concentration was set to 300 mM to weaken the electrostatic interactions between the SMC complex and DNA and accelerate DNA dynamics.

*Figure 5A, D*, *Figure 5—figure supplement 2A, D*, and *Video 4* illustrate a representative trajectory exhibiting DNA translocation by the SMC complex through one entire ATP hydrolysis cycle. *Figure 5I* represents the one-dimensional contour coordinate of the DNA molecule, indexed by base pairs (1–800). In this plot, translocation is visualized as a discontinuous shift in the range of base-pair indices that the SMC complex contracts over one complete ATP cycle. In the initial disengaged state (*Figure 5A*, *Figure 5—figure supplement 2A*), the coiled-coil arms and ATPase head domains were juxtaposed throughout the entire $1 \times 10^8$ MD steps within this state, while the DNA remained entrapped in the kleisin ring. Switching the reference potential to that of the engaged state due to ATP binding rapidly induced the transition of the SMC conformation. After some lag time, we observed a gradual tilting of DNA along the side surface of the ATPase heads (*Figure 5B*, left, and *Figure 5—figure supplement 2B*, left); then, as DNA progressively formed hydrogen bonds to the top surface of the ATPase heads during the time from $1 \times 10^8$ to $3 \times 10^8$ MD steps (*Figure 5—figure supplement 2E*), a downstream segment of DNA was captured into the hinge on the opposite side of the SMC complex (*Figure 5B*, right, *Figure 5—figure supplement 2B*, right, and *Figure 5I*), as predicted by the segment capture model. This process formed a DNA loop structure that was stabilized on one end by binding to the SMC heads and on the other by binding to the hinge. In the subsequent V-shape state adopted upon ATP hydrolysis, the SMC complex maintained the DNA loop structure within the SMC ring (*Figure 5C* and *Figure 5—figure supplement 2C*). While the ATPase heads and kleisin kept their contact with DNA, the DNA occasionally detached from the hinge domain due to the weak binding affinity. Transitioning from the V-shape to disengaged states upon ADP unbinding (*Figure 5D*, and *Figure 5—figure supplement 2D*), the DNA was pushed from the hinge domain toward the kleisin ring as the SMC complex closed its coiled-coil arms. The DNA segment initially bound to the kleisin ring detached, and the opposite DNA segment pumped from the hinge domain was instead entrapped in the kleisin ring. Consequently, the SMC complex achieved the DNA translocation, returning to its original configuration but with the DNA shifted by the size of the loop entrapped during segment capture. This translocation is recorded in *Figure 5I* as the average coordinate of the kleisin contact region (red dots) jumps from ~400 bp before the cycle to ~600 bp after, which corresponds to a translocation event of ~200 bp.

Notably, *Figure 5E–H, J*, *Figure 5—figure supplement 3*, and *Video 5* demonstrate that the SMC complex can translocate along DNA even when the SMC hinge did not capture DNA in the engaged state (*Figure 5F*, right, and *Figure 5—figure supplement 3B*). This indicates that DNA binding to the hinge domain is not essential for translocation as long as the DNA loop is captured within the SMC ring, explaining previous findings (*Bürmann et al., 2017*; *Nomidis et al., 2022*; *Vazquez Nunez et al., 2019*).

The average and standard deviation of the step size observed in the successful translocation was 216 ± 71 bp (ranging from 42 to 356 bp, *Figure 5K*, *Figure 5—figure supplement 4*), close to the length of the DNA loop captured within the SMC ring in the engaged state, 206 ± 30 bp (*Figure 5L*). The variation of step size and the occasional discrepancy between the step and loop sizes originates from the sliding of DNA around the ATPase heads, hinge, and coiled-coil arms through the

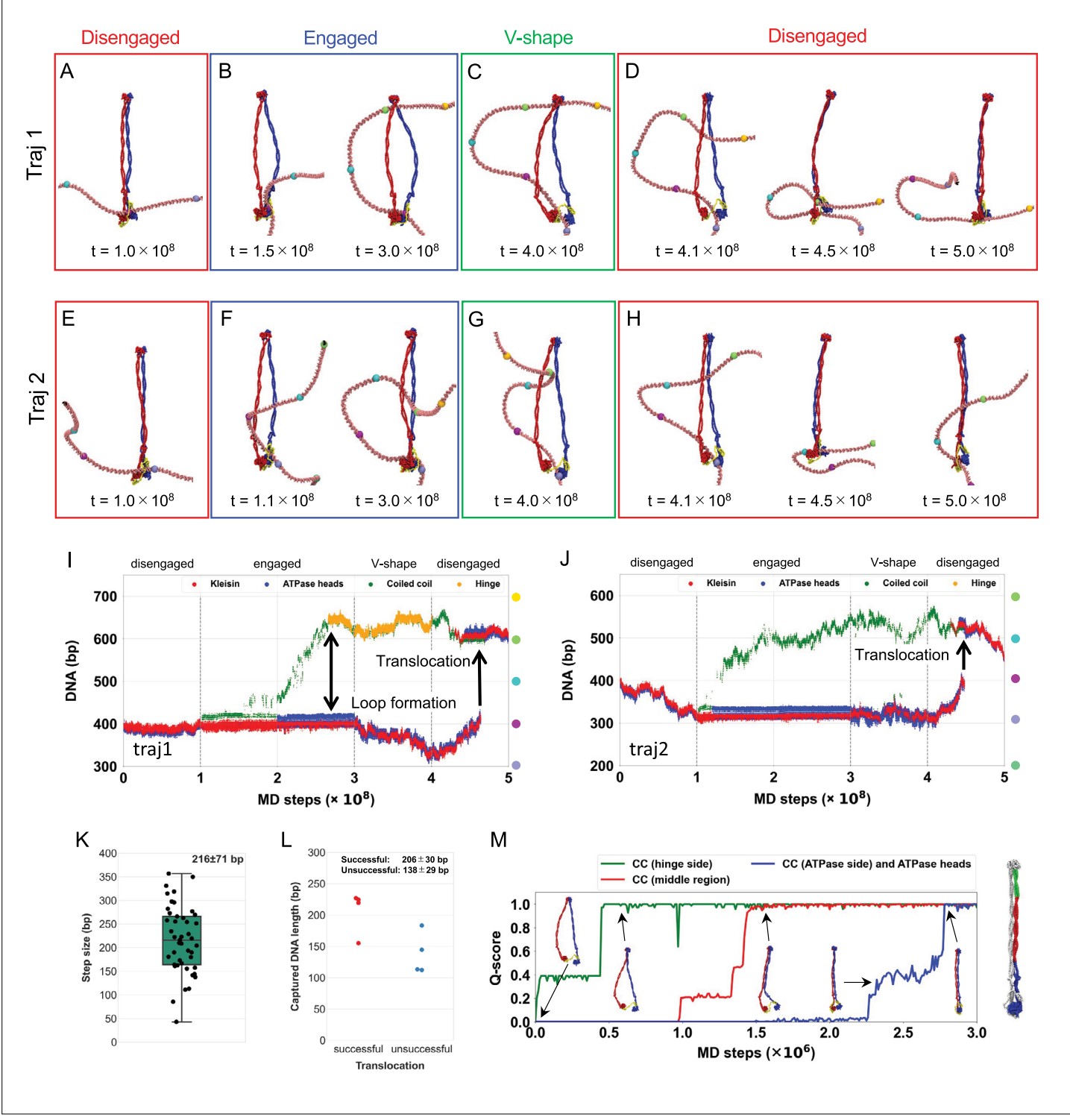

**Figure 5.** SMC translocation along DNA via DNA-segment capture. (**A–D**) A representative trajectory of DNA translocation by the SMC–ScpA complex coupled with the conformational change depending on the nucleotide states. The DNA reaches the hinge domain in the engaged state. (**E–H**) A representative trajectory of DNA translocation by the SMC–ScpA complex coupled with the conformational change depending on the nucleotide states. The DNA does not reach the hinge in the engaged state. (**I, J**) Time series of the DNA position where each SMC domain contacts with. DNA-Protein contacts at kleisin, ATPase heads, coiled-coil, and hinge domains are plotted in red, blue, green, and orange, respectively. (**K**) Analysis of translocation step size. (**L**) The length of the captured DNA segment in the engaged state for successful (left) and unsuccessful (right) translocation trajectories. (**M**) *Q*-scores, that is, the fraction of native contacts, between the intermolecular coiled-coil arm and ATPase head domains, revealing the zipping motion of

*Figure 5 continued on next page*

*Figure 5 continued*

the coiled-coil arm when transitioning from the V-shape to the disengaged state. The coiled-coil arm was divided into three domains: hinge side (green), middle region (red), and ATPase heads side (blue).

The online version of this article includes the following figure supplement(s) for figure 5:

**Figure supplement 1.** An initial structure of DNA translocation simulations.

**Figure supplement 2.** Detail snapshots during DNA translocation via DNA-segment capture.

**Figure supplement 3.** Detail snapshots during DNA translocation via DNA-segment capture where the DNA does not reach the hinge domain in the engaged state.

**Figure supplement 4.** Time series of the DNA position where each SMC and kleisin domain contacts.

**Figure supplement 5.** Translocation step size for (**A**) trajectories in which the DNA reaches the hinge domain and for (**B**) trajectories in which the DNA does not reach the hinge in the engaged state.

**Figure supplement 6.** Probability distribution of (**A**) electrostatic and (**B**) hydrogen-bonding energies between the SMC complex and DNA.

**Figure supplement 7.** Detail trajectories of simulations for SMC complex and DNA without hydrogen-bonding interactions between SMC ATPase heads and DNA.

**Figure supplement 8.** Zipping up the coiled-coil arms when transitioning from V-shape to disengaged states.

ATP-hydrolysis cycle. The step size observed when the DNA did not reach the hinge domain was 225 ± 82 bp (*Figure 5—figure supplement 5*, right), which was compatible to the step size of 207 ± 58 bp when the DNA reached the hinge domain (*Figure 5—figure supplement 5*, left).

## Hydrogen bonds between SMC ATPase heads and DNA are essential to drive DNA-segment capture

Our coarse-grained simulations of the SMC complex and DNA accounted for long-range electrostatic and short-range hydrogen-bonding interactions between the SMC ATPase head and DNA. The time series of interaction energies (*Figure 5—figure supplements 2E and 3E*) revealed that the stabilization of both electrostatic and hydrogen-bonding interactions occurs at the moment when DNA binds to the top of the ATPase heads, resulting in DNA-segment capture. *Figure 5—figure supplement 6* provides the energy distributions of electrostatic and hydrogen-bonding interactions.

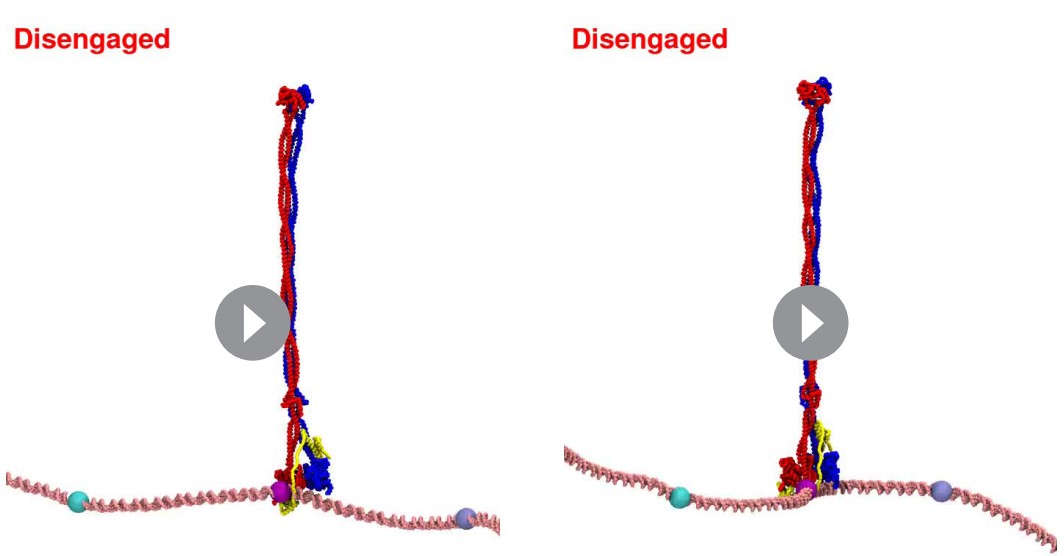

**Disengaged** **Disengaged**

**Video 4.** The representative trajectory of SMC–ScpA translocation along DNA, in which the DNA reaches the hinge domain in the engaged state.
https://elifesciences.org/articles/106752/figures#video4

**Video 5.** The representative trajectory of SMC–ScpA translocation along DNA, in which the DNA does not reach the hinge domain in the engaged state.
https://elifesciences.org/articles/106752/figures#video5

Here, we classified the engaged trajectories into those that successfully led to DNA-segment capture and those that remained in intermediate states without achieving DNA-segment capture. The results indicate that trajectories that achieve DNA-segment capture exhibit peaks at a lower energy region for electrostatic and hydrogen-bonding interactions than those remaining in the intermediate states (*Figure 5—figure supplement 6A*, top two lines, and *Figure 5—figure supplement 6B*, top two lines).

To elucidate whether electrostatic or hydrogen-bonding interaction drives DNA-segment capture, we conducted additional simulations in which hydrogen-bonding interactions were disabled. A representative trajectory from these simulations is illustrated in *Figure 5—figure supplement 7*. Across 50 simulations, no DNA-segment capture was observed, contrary to results obtained from simulations with hydrogen-bonding interactions. The absence of DNA-segment capture motion is further supported by the disappearance of the peak at the lower energy region, which is characteristic of segment capture, from the energy distributions of the electrostatic interaction (*Figure 5—figure supplement 6A*, third and fourth lines). These findings demonstrate that hydrogen-bonding interactions between the upper surface of the ATPase head and DNA are essential for driving DNA-segment capture.

## Zipping motion of coiled-coil arms pushes the DNA from hinge domain toward kleisin ring

In the translocating trajectories, DNA consistently moved from the hinge domain toward the kleisin ring in the final disengaged state (*Figure 5D*, *Figure 5—figure supplement 2D*, and *Figure 5—figure supplement 3D*), never moving backward from the ATPase heads to the hinge domain. To elucidate the molecular mechanism underlying this phenomenon, we explored the MD of the coiled-coil arms when the SMC complex transitions from the V-shape to the disengaged state. *Figure 5M* and *Figure 5—figure supplement 8* illustrate the intermolecular Q-scores for the coiled-coil arms and the ATPase head domains. The coiled-coil arms closed sequentially, initially forming intermolecular contacts at the hinge side, followed by the middle, and finally at the ATPase head side and between the ATPase head domains (*Video 6*). Therefore, the zipping up of the coiled-coil arm consistently occurs sequentially from the hinge to the ATPase head domains, facilitating the unidirectional pushing of the captured DNA segment from the hinge toward the ATPase heads. This active, mechanical pushing of the DNA loop, driven by the sequential closing of the coiled-coil arm, constitutes the physical basis of the 'pumping' mechanism that drives unidirectional translocation. Our simulations thus provide a concrete, molecular-level visualization for this key step in the DNA-segment capture model.

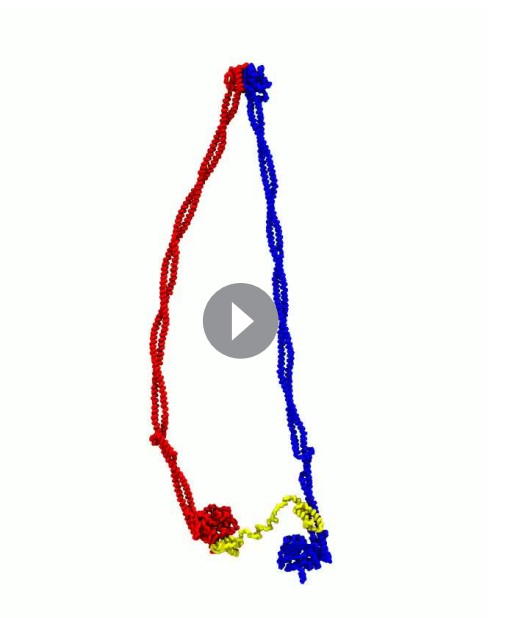

**Video 6.** The representative trajectory of zipping up the coiled-coil arm when transitioning from V-shape to disengaged states.

https://elifesciences.org/articles/106752/figures#video6

## DNA can be slipped or trapped in the coiled-coil arm during the cycle

In the above simulations, the motions of DNA during the cyclic conformational change in SMC–ScpA were highly stochastic. Only fractions of the runs (45 of 770 runs) exhibited DNA translocation, so we investigated the conditions required for productive translocation. We summarized the range of observed DNA dynamics in *Figure 6*. First, during the simulations in the disengaged state, the system displayed highly similar behavior across all 50 runs: DNA remained in the kleisin ring with some unbiased diffusion (*Figure 6A, B*). During the transition from disengaged to engaged state, the DNA bound to the top surface of the ATPase heads via hydrogen bonds, and a DNA segment was captured within the SMC ring

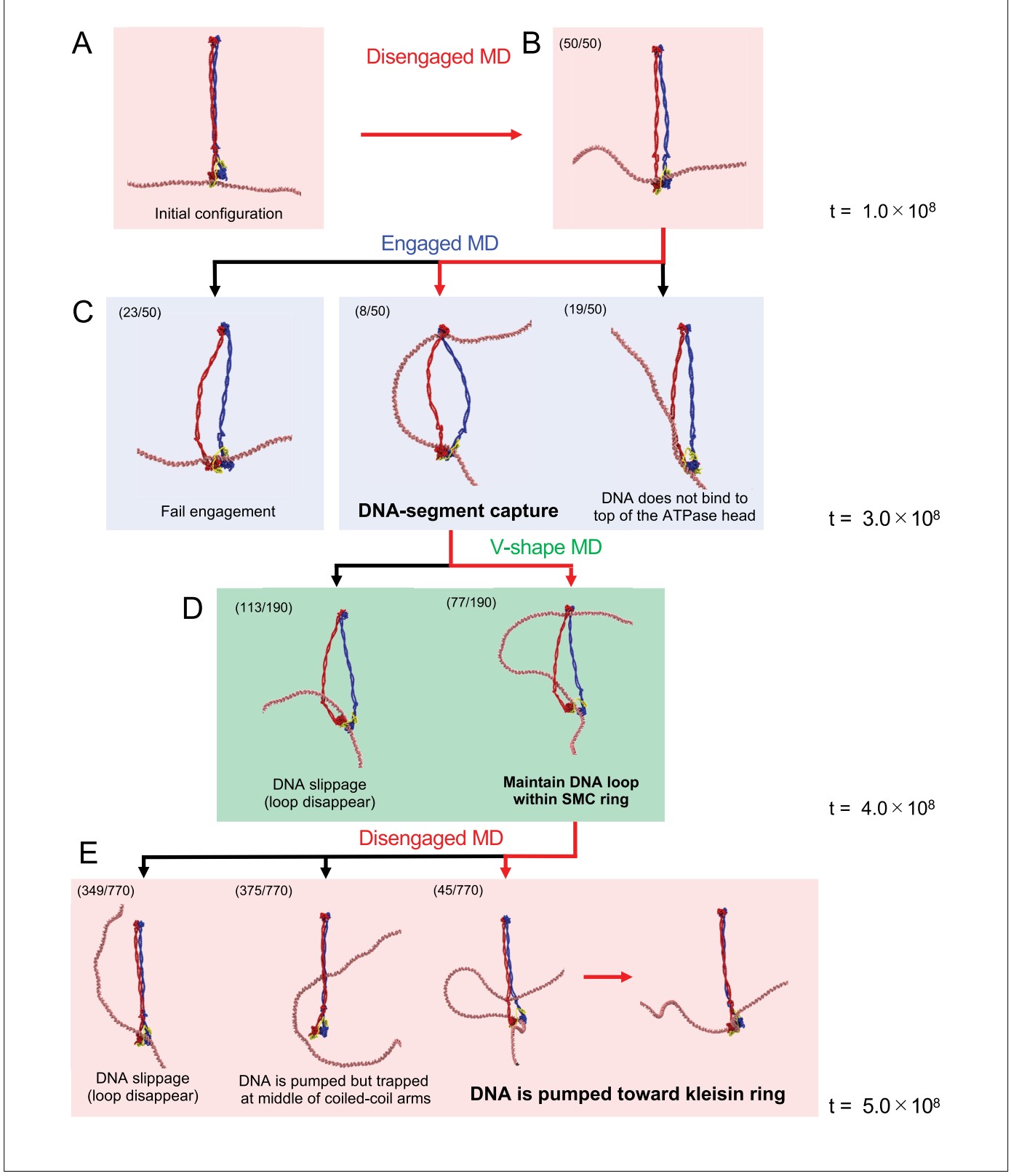

**Figure 6.** Diverse DNA dynamics during ATP hydrolysis cycle. The numbers at the top left represent the number of trajectories observed. (**A**) Initial configuration of the simulations. (**B**) The results of disengaged molecular dynamics (MD) simulations. (**C**) The results of engaged MD simulations. Each simulation was restarted from the final snapshot of each disengaged MD simulation. (**D**) The results of V-shape MD simulations. Multiple simulations

*Figure 6 continued on next page*

*Figure 6 continued*

were conducted by restarting from the DNA-segment capture trajectories in the engaged state. The results of disengaged MD simulations. The simulations were restarted from the V-shape conformations maintaining the DNA loop within the SMC ring.

The online version of this article includes the following figure supplement(s) for figure 6:

**Figure supplement 1.** Time series of the captured DNA length within the SMC ring in the engaged state during $t = 1.0 \times 10^8$ to $3.0 \times 10^8$ MD steps.

---

in 8 of 50 runs (*Figure 6C*, middle). The DNA reached the hinge domain in one of the eight runs, while DNA did not reach the hinge domain in the remaining seven runs. In some other trajectories, the SMC heads did not reach the engaged form, possibly inhibited by DNA interacting with the head domains on the side interfaces (23/50 runs) (*Figure 6C*, left). We also observed many runs in which DNA was not caught by the top surface of the ATPase heads at the end of runs (19/50 runs), and thus, the DNA loop was not formed (*Figure 6C*, right). These final structures that did not capture DNA segments resemble intermediate states of the successful DNA capture pathways (the intermediate states are observed in energy distribution of electrostatic energy in the top two lines of *Figure 5—figure supplement 6A*), indicating this is mainly due to the limited simulation time.

We simulated the transition to the V-shape state from the snapshot in which a DNA segment was captured. The DNA loop captured within the SMC ring mainly remained unchanged in 77 of 190 runs (*Figure 6D*, right), while in the remaining, it disappeared due to DNA slippage. The fundamental cause of slippage was the destabilization of the interactions between DNA and the top surface of the SMC heads upon ATP hydrolysis, as hydrogen bonds were excluded from the model due to the dissociation of the head domain dimer in the V-shape state. The DNA that was interacting with the top interface of the ATPase heads moved to the patch on the side interface of the ATPase heads. The dissociation of the two ATPase heads further destabilized the interaction with DNA, leading to DNA sliding along the hinge and coiled-coil arms, ultimately destabilizing the DNA loop. The release of the DNA loop leads to an overall configuration similar to that before the DNA-segment capture. DNA slippage could also occur in actual SMC complexes, although its probability may depend on the specific SMC and system conditions.

Starting from the structures that maintained the DNA loop in the above simulations, we conducted MD simulation of the subsequent disengaged state. In 45 of 770 runs, the captured DNA segment in the SMC ring was pumped down toward the kleisin by the zipping of the coiled-coil arms, resulting in successful DNA translocation (*Figure 6E*, right). In 349 cases, the DNA loop disappeared, and the kleisin kept binding to DNA close to the original site, similar to the DNA slippage described above (*Figure 6E*, left). We also observed trajectories where the DNA remained trapped in the middle of the coiled-coil arms during its motion from the hinge to the kleisin (*Figure 6E*, the second from the left, 375 cases). This was caused by the high strength of the interactions between the coiled-coil arms in the disengaged state, which suggests that repeated opening and closing of the coiled-coil arms is required to avoid trapping.

We finally investigated the dependence of the success rate of translocation as a function of the size of the captured DNA loop formed at the end of the engaged state. The length of the DNA loop for trajectories resulting in translocation was 206 ± 30 bp (*Figure 5L*). In contrast, the loop length for trajectories not resulting in translocation was shorter, measuring 138 ± 29 bp (*Figure 5L*), suggesting the existence of a critical loop size required for translocation. The time series analysis of the captured DNA length (*Figure 6—figure supplement 1*) indicated that a DNA loop consisting of ~100 bp tends to be formed as an intermediate state. Since the trajectories not leading to translocation were trapped in this state, more extended simulations may be necessary to form a sufficiently large loop for translocation.

## Asymmetric kleisin path makes unidirectionality of SMC translocation

In the above simulations, we consistently observed DNA translocations by the SMC complex in the same direction along the DNA molecule. Unidirectionality of DNA translocation requires breaking the symmetry of the SMC–ScpA complex. Since prokaryotic SMC proteins form a homodimer, this can only be ensured by binding to the ScpA subunit. Specifically, the ScpA NTD binds to the coiled-coil region close to the head in one SMC subunit (in blue in *Figure 2*), while the ScpA CTD binds to the bottom side of the head in the other SMC subunit (in red), imposing the symmetry breaking.

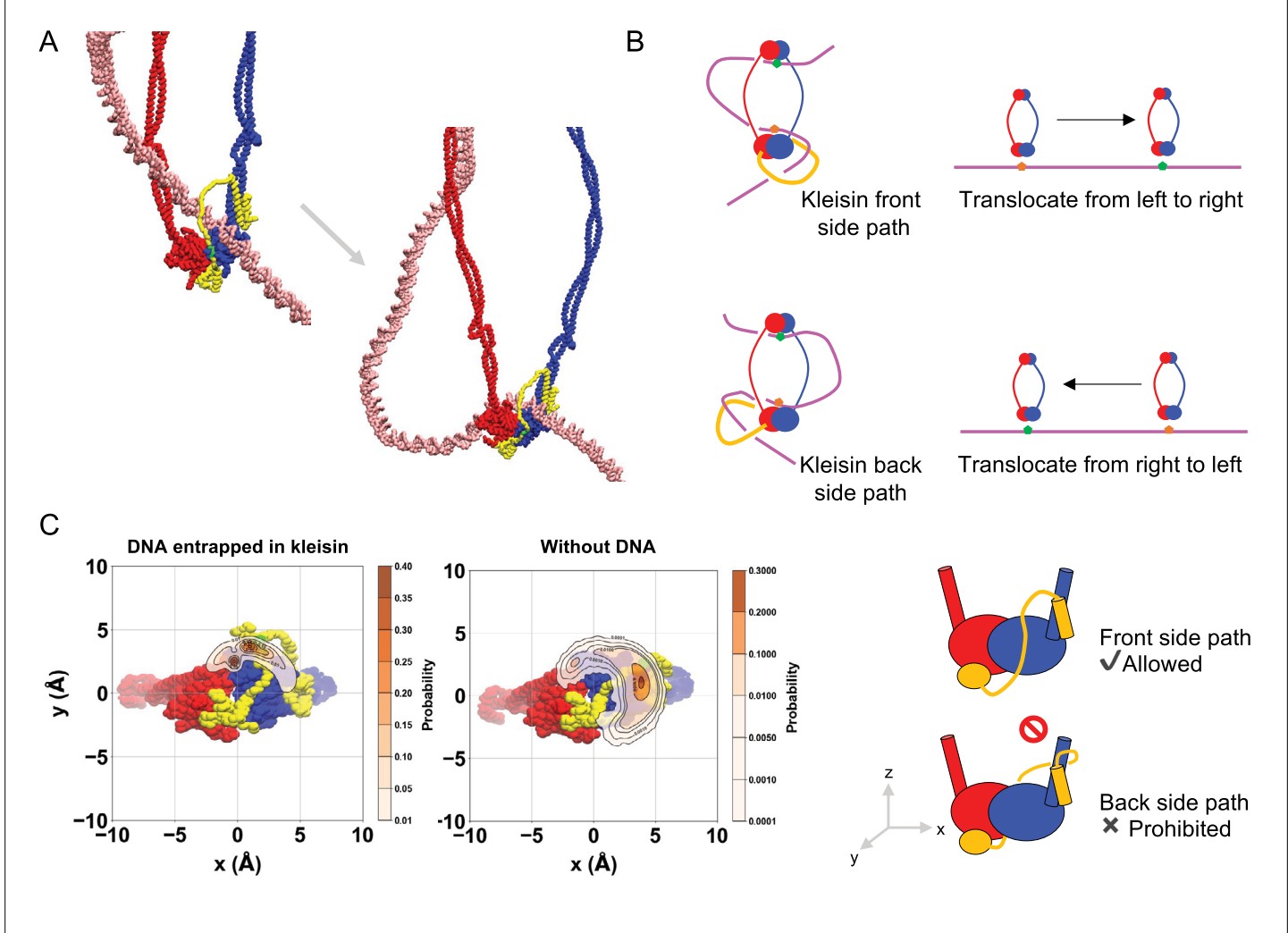

**Figure 7.** Asymmetric kleisin path makes unidirectionality of SMC translocation. (**A**) Typical snapshots of the moment when the SMC–ScpA complex captured a DNA segment within its ring structure in the engaged state. Particles marked in green on the ScpA indicate DNA patches. (**B**) Schematic figures highlighting how the kleisin path determined the direction of translocation. (**C**) The spatial distribution of the DNA patch on the ScpA subunit.

The online version of this article includes the following figure supplement(s) for figure 7:

**Figure supplement 1.** The spatial distribution of the DNA patch on the ScpA subunit that has different linker lengths.

*Figure 7A* shows typical snapshots of when the SMC–ScpA complex captured a DNA segment within its ring structure in the engaged state. The DNA loop was always inserted into the SMC ring from the side where the ScpA was located in all successful segment capture trajectories. This result led to the hypothesis that the path of ScpA (kleisin) determined the direction of translocation (*Figure 7B*). In this model, we considered two possible paths of the kleisin ring relative to the ATPase heads: the front side path and the backside path. When the kleisin traverses in front of the heads, topological constraints require a DNA loop inserted from the front side of the SMC complex (*Figure 7B*, upper left). Upon completion of SMC translocation, the DNA segment marked in orange escapes from the complex. In contrast, the segment marked in green is pumped from the hinge domain to the kleisin ring and entrapped there, resulting in the SMC translocation from left to right relative to the DNA (*Figure 7B*, upper right). On the other hand, when the kleisin traverses behind the ATPase heads, a DNA loop forms on the front side (*Figure 7B*, lower left), resulting in the SMC translocation from right to left relative to the DNA (*Figure 7B*, lower right). Thus, the kleisin path determines the direction of SMC translocation, and unidirectional SMC translocation occurs if the kleisin path is restrained on one side.

To reveal whether the path of kleisin exhibits a bias, we computed the spatial distribution of the center of mass of the ScpA DNA-binding patch (K126–K130) relative to the SMC heads in the engaged state (*Figure 7C*, left), finding this was indeed restrained exclusively to one side. This ScpA location was consistent with the side where the DNA loop was inserted into the SMC ring, as seen in *Figure 7A*. To rule out the possibility that the biased positioning of ScpA was a result of DNA–kleisin interaction, we further analyzed the spatial distribution of ScpA in the absence of DNA, finding a similar positional bias (*Figure 7C*, middle). In other words, kleisin binding restrains its configuration to a specific side of the SMC heads. This restraint was in part due to the short length of the disordered region of ScpA, consisting of 63 residues (residues 82–144 of ScpA), which structurally impedes reaching the backside, and in part to the peptide chains of ScpA emanating toward only one side of the SMC at their interface with the complex as they bind to the two SMC subunits.

To test the importance of the shortness of the kleisin loop to break the left–right symmetry of the prokaryotic SMC dimers, we constructed SMC–ScpA models in which the ScpA was elongated by introducing multiples of [-GGGGS-]$_x$ linkers upstream and downstream of the ScpA DNA-binding patch (K126–K130) ($x$ = 0–10, *Figure 7—figure supplement 1A*). The spatial distribution of DNA patches observed upon the introduction of GGGGS linkers is shown in *Figure 7—figure supplement 1B*. While the addition of short linkers ($x$ = 2, 4, 6) in ScpA left the asymmetric path as in the wild type, the longer linkers ($x$ = 8, 10) abolished the asymmetry of the ScpA path because the elongated ScpA was able to reach the opposite side of the ATPase head, demonstrating that the longer linkers break the asymmetry of the SMC dimer.

In summary, we revealed that the asymmetric ScpA path is the critical determinant of the SMC unidirectional DNA translocation via a DNA-segment capture.

## Discussion

### DNA-segment capture as the mechanism of the DNA translocation by SMC complexes

DNA translocation and loop extrusion by SMC complex have been hypothesized to be the fundamental mechanisms for chromosome compaction and the formation of topologically associating domains (*Alipour and Marko, 2012*; *Fudenberg et al., 2016*; *Goloborodko et al., 2016a*; *Goloborodko et al., 2016b*; *Nasmyth, 2001*). Progress has been made in the past decade toward elucidating the molecular mechanisms behind the motor activity of the SMC protein (*Davidson and Peters, 2021*; *Higashi and Uhlmann, 2022*; *Kim et al., 2023*). However, the relationship between ATP-dependent conformational changes occurring in SMC complexes and translocation along DNA has not been fully understood, and the proposed mechanisms *Davidson and Peters, 2021*; *Higashi and Uhlmann, 2022*; *Kim et al., 2023* have yet to be thoroughly tested at the molecular level.

In this study, we conducted all-atom and residue-resolution MD simulations to elucidate the molecular mechanism underlying DNA translocation by SMC complexes. To simplify the system, this study examined the prokaryotic SMC complex consisting of the SMC homodimer and kleisin molecule ScpA. Notably, we employed a bottom-up modeling approach, allowing the mechanism of SMC action to emerge from the simulations without assuming any particular model of DNA translocation. Utilizing structure-based residue-resolution coarse-grained simulations, we successfully reproduced the ATP-dependent cyclic conformational change of the SMC–ScpA complex (*Figure 3*). The structural features of the SMC complex, such as rod shape in the disengaged (Apo) state, open ring in the engaged (ATP-bound) state, and V-shape-like structure in the V-shape (ADP-bound) state shown in *Figure 3A, B* are in good agreement with experimental observations by electron micrographs (*Kamada et al., 2017*). We then identified DNA-binding sites at the top of the ATPase heads, the inner side of the hinge, the middle of the coiled-coil arms, and the ScpA disordered region within the full-length SMC complex in the ATP-bound engaged state (*Figure 4*). These DNA-binding sites agree with the biochemical assays (*Griese and Hopfner, 2011*; *Hirano and Hirano, 2006*; *Vazquez Nunez et al., 2019*). Extensive simulations with a long duplex DNA entrapped in the SMC–ScpA complex throughout cyclic conformational changes revealed that DNA translocation proceeds via the segment capture mechanism: the complex captures a DNA segment in the engaged state to form a DNA loop structure within the SMC ring, and the segment is subsequently pumped toward the kleisin ring, leading to DNA translocation (*Figure 5*). We demonstrated that the hydrogen-bond interactions between the SMC ATPase

heads and DNA are essential to drive the DNA-segment capture (*Figure 5—figure supplement 6*). Using all-atom simulations, we identified critical amino acid residues such as Arg120, Arg123, Arg111, Arg62, and Lys56 (*Figure 1B*, *Figure 1—figure supplement 2B*). Based on these findings, we propose future experiments that mutate these basic residues to inhibit DNA-segment capture and abolish translocation activity. We also demonstrated that the hinge-DNA interaction in the engaged state is not necessary for DNA translocation via DNA-segment capture (*Figure 5—figure supplement 3*), which explains previous experimental findings (*Bürmann et al., 2017*; *Nomidis et al., 2022*; *Vazquez Nunez et al., 2019*).

The step size, the number of DNA base pairs translocated per one ATP cycle, was estimated as 216 ± 71 bp (*Figure 5K*). This was largely determined by the length of the DNA loop captured in the engaged state (*Figure 5L*), which is, in turn, affected mainly by the distance between the ATPase heads and the hinge domains. Since the overall SMC architecture and size are well-conserved, a similar step size should also be expected for other SMC complexes. Our results are consistent with a previous simulation study using a coarser mesoscopic model, which suggested a step size of ~180 bp without DNA tension (*Nomidis et al., 2022*). A previous DNA-curtain experiment estimated that yeast condensin translocases along DNA with an average step size of 60 bp (*Terakawa et al., 2017*), which is a lower bound, as it assumes that all ATP hydrolysis events lead to productive translocation. Magnetic tweezer experiments have measured the median step size of yeast condensin during the loop extrusion process to be 20–40 nm, corresponding to 60–200 bp per step, at DNA stretching forces from 1 to 0.2 pN (*Ryu et al., 2022*). The loop extrusion speed and ATPase rates were also estimated in in vitro studies as ~0.5–1 kbp/s and 2 ATP/s for condensin (*Ganji et al., 2018*), cohesin (*Davidson et al., 2019*; *Kim et al., 2019*), and SMC5/6 (*Pradhan et al., 2023*) and 300–800 bp/s and 1 ATP/s for *B. subtilis* (*Wang et al., 2018*; *Wang et al., 2017*) and *C. crescentus* (*Tran et al., 2017*) SMCs. A theory also estimated loop extrusion speed as ~0.5 kbp/s without tension and a maximum of ~1.5 kbp/s with tension (*Takaki et al., 2021*). These step sizes are somewhat larger than, but still compatible with our estimates.

Recent magnetic tweezer experiments revealed a DNA twist of −0.6 at each DNA-loop extrusion step for all eukaryotic SMC proteins (*Janissen et al., 2024*), implying a common DNA-loop extrusion mechanism. While it would be interesting to elucidate the molecular mechanism of the twist, this is challenging due to important differences between the magnetic tweezer experiments and the current coarse-grained simulation setup. First, we addressed DNA translocation by the SMC complex, but not the DNA-loop extrusion in this study. A safety belt, which fixes the DNA at a certain location on the complex, may be essential to convert translocation mode to a DNA-loop extrusion mode and generate twists. However, the current simulation model did not incorporate a possible safety belt such as KITE and HAWK subunits. In the simulation, the DNA ends are also free to move, releasing any twist accumulated during SMC activity. The flexibility of the coiled-coil arms may also explain the twist of −0.6 upon completion of the DNA-loop extrusion step. In the current coarse-grained simulation, the coiled-coil arms are modeled based on the reference structure of the rod-shaped SMC complex, potentially making them slightly stiffer than in reality. Therefore, we anticipate that a different simulation setup is necessary to examine a twist change of −0.6 at each DNA-loop-extrusion step.

Folding of the SMC coiled-coils at so-called elbows has been shown in the condensin (*Lee et al., 2020*), cohesin (*Bürmann et al., 2019*; *Petela et al., 2021*), and MukBEF (*Bürmann et al., 2021*; *Bürmann et al., 2019*), with some models suggesting its role in DNA translocation (*Bauer et al., 2021*; *Davidson and Peters, 2021*; *Higashi et al., 2021*; *Higashi and Uhlmann, 2022*; *Kim et al., 2023*; *Ryu et al., 2022*; *Ryu et al., 2020*). However, since evidence of elbow folding is absent in prokaryotic SMC (*Diebold-Durand et al., 2017*; as well as in SMC5/6; *Hallett et al., 2022*; *Yu et al., 2021*), this was not incorporated into our coarse-grained model. Our simulations realized the DNA translocation via DNA-segment capture without assuming elbow folding, suggesting elbow folding is not strictly required for SMC activity. The previous mesoscopic simulation based on the segment capture model (*Nomidis et al., 2022*) showed that making flexible elbows reduces the step size of translocation, implying that elbow folding may hinder rather than facilitate DNA translocation. The 'hold-and-feed' mechanism, a revised DNA-segment capture model recently proposed based on experiments (*Shaltiel et al., 2022*), also places more emphasis on the motion of DNA through the chambers formed by the kleisin and HAWK subunits, rather than on elbow folding, without strictly ruling out elbow folding motion.

Turning to loop extrusion mechanisms, alternative mechanisms have been proposed in addition to the DNA-segment capture model. For example, Takaki et al. developed a scrunching-based theory that quantitatively accounts for several experimental observations, including force-velocity relationships and step-size distributions. While our present study focuses on the DNA translocation mechanism via segment capture, it is important to note that scrunching and other models remain plausible alternatives for loop extrusion. The precise mechanism may depend on the specific SMC complex and their subunits and remains to be fully resolved.

## Unidirectionality of DNA translocation and DNA-loop extrusion

Our simulations demonstrated unidirectional DNA translocation by the SMC–ScpA complex. The unidirectionality is realized because the ScpA conformation exhibits a one-sided bias relative to the ATPase heads (*Figure 7C*), causing the DNA loop to be consistently captured into the SMC ring from the side where the ScpA is located (*Figure 7A*). The one-sided biased path of ScpA originates from the short length of the ScpA disordered region and the fact that the peptide chains of ScpA emanate toward only one side of the SMC at their interface with the complex as they bind to the two SMC subunits.

Our simulations suggest that the shortness of the kleisin loop is critical to break the left–right symmetry of the prokaryotic SMC homodimer. Elongated ScpA reduced or abolished the asymmetry, enabling it to reach the opposite side of the ATPase head (*Figure 7—figure supplement 1B*), which would lead to malfunction in the unidirectional translocation of the prokaryotic SMC complex. Future experiments can test this.

On the other hand, eukaryotic SMC complexes, including condensin and cohesin, possess longer kleisin subunits that may not efficiently break the symmetry due to their ability to locate both on the front and the back sides of the ATPase heads (*Figure 7—figure supplement 1B*). Inspired by the DNA-clamping state (*Bürmann et al., 2021*; *Collier et al., 2020*; *Lee et al., 2022*; *Shi et al., 2020*; *Yu et al., 2022*), where the KITE and HAWK subunits interact with the SMC ATPase heads from a specific direction, we propose that these subunits may be the key factors to break the left–right symmetry in eukaryotic SMC complexes. The 'hold-and-feed' model for condensin's DNA-loop extrusion further supports this hypothesis (*Shaltiel et al., 2022*). In this model, a DNA loop is entrapped in two separate kleisin chambers: a chamber created by the kleisin and Ycg1 anchors a DNA segment. In contrast, another chamber created by the Ycs4 and kleisin provides the motor function through a power stroke from one side when the condensin transits from the ATP-free state to the ATP-bound state in gripping conformation. The Ycs4 subunit breaks the left–right symmetry and ensures the unidirectionality of the DNA-loop extrusion. Further studies at the molecular level focused on the role of the kleisin and the accessory SMC subunits will be of great importance to elucidate within a unified view how SMC complexes realize their unidirectionality during DNA translocation and loop extrusion.

## Parametric choices and robustness of simulation model

The switching Gō approach adopted in this study is a powerful tool for providing the relationship between known large-scale conformational changes and the resulting functional and mechanical dynamics of the molecular machine (*Brandani and Takada, 2018*; *Koga and Takada, 2006*; *Nagae et al., 2025*). In this study, we mimic conformational change induced by ATP-binding and hydrolysis events by instantaneously switching the potential energy function from one that stabilized a given conformation to another that stabilized a different conformation. This drives the protein to undergo a conformational transition toward the minimum of the new energy landscape.

This approach is particularly well-suited to investigate whether a given conformational change in a subunit of a molecular machine can produce the overall motion observed, and whether this process is mechanically feasible. Therefore, the fundamental mechanisms identified in this study, that is, DNA-segment capture mechanism, the correlation between step size and loop length, and the unidirectional translocation mechanism originating from the asymmetric kleisin path, can be considered as robust, as they emerge directly from the structural and topological constraints of the SMC–kleisin architecture rather than from tuned parameters.

On the other hand, the efficiency and success rate of DNA translocation in our simulations are more sensitive to certain parametric choices. For instance, the selection and strength of hydrogen-bond-like interactions are a key factor. Our model incorporates specific hydrogen bonds between the

upper surface of the ATPase heads and DNA, based on all-atom simulations. These interactions are essential for initiating segment capture; without them, DNA fails to migrate to the correct binding surface. While the identification of these key residues is a robust finding—persisting across different all-atom force fields (*Figure 1—figure supplement 2*)—their strength and number in the coarse-grained potential are critical parameters that directly influence the probability and kinetics of DNA capture. Another critical parameter is the ionic strength. We performed translocation simulations at an ionic strength of 300 mM to accelerate DNA dynamics. At lower concentrations, non-specific electrostatic interactions between DNA and positively charged patches on the sides of the ATPase heads or coiled-coil arm became dominant, hindering the efficient migration of DNA to its functional binding site. Using a higher-than-physiological ionic strength is a justified practice in coarse-grained simulations employing the Debye–Hückel approximation, as it serves as a first-order correction to mimic the strong local charge screening by condensed counterions that is not explicitly captured by the mean-field model (*Brandani et al., 2021*). Finally, the interaction strength between the coiled-coil arms is also important. In our model, once the arms closed during the transition from the V-shaped to the disengaged state, they remained closed on the simulated time scale, frequently trapping DNA pushed from the hinge and thereby leading to failed translocation. This behavior suggests that the arm–arm interactions may be overestimated. A parameterization that allows for more frequent, transient opening of the arms could increase the success rate of DNA pumping.

## Limitations in current simulations

Here, we discuss the limitations and possible future improvements of our model, which is, to our knowledge, the first to reveal the molecular mechanism of DNA translocation by SMC complexes at residue resolution.

First, the use of switching potentials to trigger conformational changes imposes a limitation on predictive power for energetics and transition pathways. The switching of potentials is akin to a 'vertical excitation' from one energy landscape to another, rather than a thermally activated crossing of an energy barrier. Consequently, the model cannot provide quantitative predictions of the transition rates or the free-energy barriers associated with these changes. Furthermore, while the subsequent relaxation follows the new potential landscape, it is not guaranteed to reproduce the unique, physically correct transition pathway. Nevertheless, this simplification is justified because conformational changes within the protein are expected to occur on a much faster time scale than the large-scale motion of the DNA. Thus, this simplification has a limited impact on our main conclusions regarding the functional DNA dynamics driven by these large-scale conformational changes.

Second, the time scale that is reachable by the simulations is one of the major limitations. The ATP hydrolysis rates measured for SMC complexes range between 0.1 and 2 s$^{-1}$ (*Hassler et al., 2018*), orders of magnitude lower than those in typical molecular motors. While mapping our simulation time scale to the real-time is not rigorously decided, 1 MD step is effectively on the order of 1 ps, meaning our simulations cover the sub-millisecond time scale. Thus, in reality, SMC complexes sample a much broader conformational space than we could in the current simulations. Especially, the DNA-segment capture in the engaged state was the most time-consuming process in our simulations, and only a limited number of trajectories had sufficient time to fully realize this transition. The trajectories in the engaged state were often trapped in some intermediate state along the DNA capture pathway (*Figure 6C*, *Figure 6—figure supplement 1B*), inevitably compromising the subsequent DNA translocation steps during the ATP cycle. Therefore, the success rate would most likely increase if we run the simulation long enough. Also, given the ATP hydrolysis rate and the step size per second observed in in vitro experiments (*Hassler et al., 2018*), the success rate of actual DNA-segment capture must be higher than the current coarse-grained simulations. Future studies can leverage the insights from this work to overcome the current time scale limitations. Techniques such as Markov state modeling (*Husic and Pande, 2018*; *Prinz et al., 2011*) or enhanced sampling methods (*Hénin et al., 2022*) may be employed to quantitatively characterize the free-energy landscape and transition rates. Such an approach would provide a rigorous understanding of the kinetic barriers, such as the stability of the trapped state, that govern the efficiency of SMC translocation.

Related to the time scale issue, once the coiled-coil arm closed in the disengaged state, it never reopened during our simulations, frequently causing DNA to remain trapped between the coiled-coil arms as DNA was being pushed from the hinge toward the kleisin (*Figure 6E*, the second from the

left). Longer time scales could alleviate this problem but could also be an artifact derived from our current modeling choices. Modifying the interactions between the coiled-coil arms by calibrating the intermolecular interaction parameter and employing a multiple-basin potential (*Okazaki et al., 2006*) could enable repeated opening and closing of the arms, potentially improving the success rate of DNA pumping and translocation.

Third, in the current coarse-grained simulations, the efficiency and success rate of translocation could be sensitive to salt concentration, that is, the strength of electrostatic interaction between SMC and DNA. For example, we observed strong DNA patches on the side surfaces of the SMC ATPase heads (*Figure 4A, B*). These DNA patches prevent the efficient migration of DNA onto the top of the ATPase heads in the engaged state when the salt concentration is lower, reducing segment capture efficiency. Also, the SMC coiled-coil arms have a DNA patch at their middle region. This DNA patch could trap the DNA at the middle region of the coiled-coil arms when the DNA is pumped from the hinge to the kleisin ring in the last disengaged state when the salt concentration is lower. For this reason, this study used an ion concentration of 300 mM in the translocation simulations to accelerate DNA dynamics. It is worth mentioning that *P. yayanosii*, the prokaryote used in this study, is grown under 2.5–5.5% (wt/vol) NaCl conditions (optimum 3.5 %), which corresponds to ion concentrations of 420–940 mM (optimum 590 mM) (*Birrien et al., 2011*). Therefore, the 300 mM ion concentration in the translocation simulations is more physiologically relevant than the standard physiological 150 mM ion concentration. However, a lower concentration is helpful to efficiently investigate the DNA-binding sites of the SMC complex.

Fourth, in the absence of detailed knowledge about the time scales of the ATP hydrolysis, the length of each state during the cyclic transitions was prefixed in the current simulation. In reality, however, these transitions occur stochastically, determined by the rates of ATP binding, ATP hydrolysis, and ADP dissociation, and are likely dependent on the system conformation. Our simulations suggest that careful coordination between the movement of DNA through the SMC and the timing of the transitions between the ATP states is critical at multiple stages of the process. Specifically, we found that if ATP hydrolysis triggers the transition from the engaged to the V-shape state before the DNA segment is captured to form a loop, then DNA translocation cannot occur. Since DNA-segment capture was the most time-consuming conformational transition of the overall process in our simulations, the slow ATP hydrolysis rate of the ATPase (*Hassler et al., 2018*), a common feature of SMC complexes, may precisely serve the purpose of allowing a sufficient time for the formation of a DNA loop within the SMC ring. The experimentally observed DNA-stimulated ATPase activity of SMC (*Lammens et al., 2004*; *Terakawa et al., 2017*) may further contribute to coordinating the various steps of the process by ensuring ATP hydrolysis preferentially occurs when the DNA loop is formed. We also found that excessive residence time in the V-shape state can increase the risk of DNA slippage and failure of translocation (*Figure 6D*, left), indicating that a high rate of ADP dissociation may increase the efficiency of the process.

Finally, our simulations considered idealized conditions (SMC translocation on naked DNA) compared to a natural cell environment. Therefore, further investigation is needed to determine whether the translocation process depicted in this study also occurs in cells. Toward this, it is essential to reveal what happens from a molecular point of view when the SMC complexes encounter obstacles on DNA, such as RNA polymerase (*Brandão et al., 2019*; *Pradhan et al., 2022*), chromatin proteins (*Pradhan et al., 2022*; *Yamaura et al., 2024*), transcription factors, and other SMC complexes (*Brandão et al., 2021*; *Kim et al., 2020*).

## Materials and methods

### All-atom MD simulations

An initial structure of the engaged *Pf*SMC ATPase head dimer was prepared based on the crystal structure (PDB code: 1xex) (*Lammens et al., 2004*). The E1098Q mutations were modified to wild type. Missing residues and missing C-terminal regions are reconstructed with Modeller 10.1 (*Webb and Sali, 2016*). Water molecules, magnesium ions, and ATP molecules observed in the crystal structure were kept in the simulations. The ATPase domain of the SMC proteins is composed of an N-lobe and a C-lobe region, and the C-terminal of the N-lobe and the N-terminal of the C-lobe are originally connected to the missing coiled-coil arm. To remove effects originating from the terminal charges at

these termini, we attached Nme and Ace groups to the C-terminal of the N-lobe and the N-terminal of the C-lobe, respectively.

Since there is no available structure of ATPase–DNA complex for prokaryotic SMCs, we referred to a crystal complex structure of ATPγS–Mre11–Rad50 (a homolog of SMC proteins) and dsDNA complex (PDB code: 5dny) (*Liu et al., 2016*). An initial structure of the *Pf*SMC ATPase head-dsDNA complex was modeled following the previous literature (*Vazquez Nunez et al., 2019*): (1) The *Pf*SMC ATPase head dimer in the engaged state (PDB code: 1xex) was superimposed onto the ATPase domain in the Rad50–dsDNA complex (PDB code: 5dny). (2) A target dsDNA was then superimposed onto the dsDNA in the Rad50–dsDNA complex. Here, a 47-bp dsDNA (5′-GGCGA CGTGA TCACC AGATG ATGCT AGATG CTTTC CGAAG AGAGA GC-3′) (*Lammens et al., 2004*) was prepared with x3DNA 2.4 (*Lu, 2003*; *Lu and Olson, 2008*). (3) The superimposed dsDNA was slightly shifted to prevent atomic clashes between the SMC ATPase and dsDNA. Arg and Lys side chains were set as protonated, while Asp and Glu side chains were set as deprotonated. $Na^+$ and $Cl^-$ ions and solvent water molecules were distributed in a rectangular simulation box using AmberTools20 (*Case et al., 2020*). The number of ions was chosen to give an ion concentration of 150 mM.

All MD simulations were performed with OpenMM 7.7.0 (*Eastman et al., 2017*). The following force fields were adopted: AMBER ff19SB for proteins (*Tian et al., 2020*); OL15 for DNA molecules (*Zgarbová et al., 2015*); OPC four-point rigid water model (*Izadi et al., 2014*); and Carlson parameters for the ATP molecules (*Meagher et al., 2003*). We find that the choice of all-atom force field does not significantly impact hydrogen-bond formation between SMC ATPase heads and DNA (*Figure 1—figure supplement 2*), as will be discussed in detail in a later section. Coulomb interactions were calculated using the particle mesh Ewald (PME) method (*Darden et al., 1993*; *Essmann et al., 1995*). The cutoff distance for nonbonded interactions (i.e., Lennard–Jones interaction and direct space term of the PME) was set to 1 nm. Bonds involving hydrogen atoms were constrained using SETTLE (*Miyamoto and Kollman, 1992*) for water molecules and SHAKE (*Ryckaert et al., 1977*) for proteins, DNA, and ATP molecules. The temperature was controlled at 300 K using a Langevin middle integrator with a friction coefficient of 1.0 $ps^{-1}$ (*Zhang et al., 2019*), and the pressure was controlled at 1 bar with a Monte Carlo barostat (*Åqvist et al., 2004*; *Chow and Ferguson, 1995*). The time step was set to 4.0 fs by adopting a hydrogen-mass repartitioning scheme with a scale factor of 3 for each hydrogen atom (*Hopkins et al., 2015*). Periodic boundary conditions were applied throughout the simulations. Five 1 μs production runs were performed after energy minimization with the L-BFGS algorithm (*Liu and Nocedal, 1989*) and relaxation simulations with harmonic restraints on the heavy atoms. Configurations were stored every 10 ps. MDTraj (version 1.9.6) (*McGibbon et al., 2015*) was used to post-process trajectory data and analysis.

## Definition of hydrogen bonds in all-atom trajectories

Hydrogen bonds were identified based on Baker–Hubbard criteria (*Baker and Hubbard, 1984*), where the distance and angle between donor-hydrogen and acceptor satisfy the conditions $r_{H-Acceptor} < 2.5$ Å and $\theta_{DHA} > 120$, respectively.

## Modeling the full-length SMC–ScpA complex in the disengaged form

We considered the full-length *Py*SMC homodimer model (*Figure 2A*), which was previously constructed based on protein cross-linking experiments and crystal structures of individual domains (*Diebold-Durand et al., 2017*), as a reference structure. We added the ScpA subunit to this model using Modeller 10.1 (*Webb and Sali, 2016*) based on the following additional template structures: (1) the crystal structure of the *Bs*SMC ATPase head domain with the extended coiled-coil bound to NTD of ScpA adopting an extended helix structure (PDB code: 3zgx) (*Figure 2B*; *Bürmann et al., 2013*); (2) an AlphaFold2-predicted structure (*Jumper et al., 2021*; *Mirdita et al., 2022*) of the *Py*SMC ATPase head with the extended coiled-coil bound to the NTD of ScpA adopting a helix-bundle structure (*Figure 2C*); and (3) the crystal structure of the *Pf*SMC ATPase head with the extended coiled-coil bound to the CTD of ScpA (PDB code: 5xns) (*Figure 2D*; *Diebold-Durand et al., 2017*).

The ScpA NTD in the first template structure (*Figure 2B*) adopts an extended helix conformation and forms a three-helix bundle structure with the coiled-coil of SMC. However, this extended helix is likely a result of crystal packing artifacts induced by domain-swapping and is thermodynamically unfavorable due to exposing hydrophobic residues when the SMC–ScpA complex exists as a monomer

(*Kamada et al., 2017*). On the other hand, cross-link experiments for a prokaryotic *Bs*SMC complex support the four-helix bundle structure in the interface of the ScpA NTD and the SMC coiled-coil (*Gligoris et al., 2014*; *Kamada et al., 2017*), consistent with the structure predicted by AlphaFold2 (*Figure 2C*). Therefore, the extended part of the ScpA N-terminal helix derived from the first template was excluded in the modeling of the entire SMC–ScpA complex. We regarded a middle region of ScpA as a flexible linker, as it was absent in the template structures. The obtained full-length model of SMC–ScpA in the disengaged form is depicted in *Figure 2E*.

## Coarse-grained simulations

We utilized residue-resolution molecular modeling to reproduce the ATP-dependent conformational changes of the SMC–ScpA complex and observe DNA movement coupled with these conformational changes. For the SMC–ScpA complex, we adopted the AICG2+ structure-based model (*Li et al., 2014*), where each amino acid residue was represented by one bead located on Cα atom position. For DNA, we adopted the 3SPN2.C sequence-dependent model (*Freeman et al., 2014*), where each nucleotide was represented by three beads located at the positions of base, sugar, and phosphate units (*Hyeon and Thirumalai, 2005*). Electrostatic and excluded volume interactions were applied for interactions between the SMC–ScpA complex and DNA unless otherwise stated. The electrostatic interactions were modeled with the Debye–Hückel potential. The charges for globular parts of the SMC–ScpA were determined by the RESPAC algorithm (*Terakawa and Takada, 2014*) by distributing them on surface residue beads so that the electrostatic potential around the protein matches the Poisson–Boltzmann estimate based on the all-atom structure. Note that electrostatic interactions by RESPAC charges also include the contributions from salt bridge formation between the SMC ATPase heads and DNA. The standard integer charges (±1 or 0) were used for the flexible linker of ScpA in the middle region. Constant −0.6 charges were placed on the phosphate beads for intra-DNA interactions to consider counter ion condensation, although −1.0 charges were used for protein–DNA interactions.

For the disengaged state, we used the reference structure of the full-length SMC–ScpA model we have built. The intermolecular structure-based interactions between coiled-coil arms were activated only in the final disengaged state of the DNA translocation simulations and in those studying the coiled-coil zipping process in the absence of DNA. In other simulations, these interactions were turned off because the intermolecular hinge interactions and intermolecular ATPase heads interactions are sufficient to maintain the overall I-shape structure (*Figure 3B*), and to confirm that changes in the intermolecular interactions between ATPase heads alone can alter the overall structure. For the engaged state, the reference structure of the ATPase heads was based on the crystal structure (PDB code: 1xex, *Figure 2F*). The inter-subunit interaction strengths were scaled by a factor of two to accelerate transitions from the disengaged state. The reference structures for the coiled-coil arms and hinge region remain the same as those used in the disengaged state. The hydrogen-bond interactions (*Niina et al., 2017*) were introduced between the DNA and SMC ATPase heads in the engaged state. These coarse-grained hydrogen-bonding interactions use filter functions in the potential formula so that forces are applied only when the phosphate particles of DNA and the Cα atoms of the protein are within a short range and specific orientation. Note that the coarse-grained hydrogen-bonding interactions are distinguished from long-range electrostatic interactions like salt bridges because the hydrogen-bond interactions act only over short distances and with a specific directionality. The coarse-grained hydrogen bonds in our model were set up based on the all-atom simulations. In the residue-resolution model, these hydrogen bonds can be formed by the top 15 amino acid residues (fraction >0.05) from the all-atom analysis in *Figure 1B*. The rationale for this cutoff is the physical robustness of the identified interactions; all-atom simulations using a different force field confirmed that the same set of key interacting residues, including both strong and moderate binders, was consistently identified (*Figure 1—figure supplement 2*). The probability distributions of distance and angles hydrogen-bond parameters ($r$, $\theta$, and $\varphi$) were computed for those amino acid residues using one of the trajectories obtained from the all-atom simulations. The hydrogen-bond parameters were optimized according to the peak positions of the all-atom distributions (*Figure 1—figure supplement 3*). The energy constant ε was set to 4.0 $k_BT$ (~2.4 kcal/mol), within the experimental range for ideal hydrogen bonds. For the V-shape structure, we employed the same reference structure as in the disengaged state but turned off the inter-subunit interactions between the ATPase head domains, as these domains are not expected to stably interact in the V-shape state (*Hirano, 2001*; *Hirano and*

*Hirano, 2006*; *Kamada et al., 2017*; *Melby et al., 1998*). We adopted the flexible local potential (*Terakawa and Takada, 2011*) for segments between the hinge and coiled-coil arm in the engaged and V-shape states to enable an open-hinge structure.

Coarse-grained MD simulations were performed with CafeMol 3.2.1 (https://www.cafemol.org; *Kenzaki et al., 2011*) by integrating the equations of motion using Langevin dynamics with a step size of 0.2 in the CafeMol time unit. The temperature was set to 300 K.

### Analysis of DNA-segment capture and translocation

The captured DNA loop size was estimated as the difference between the average base positions of DNA in contact with ScpA and those in contact with the coiled-coil arms. We calculated the averages using snapshots from the last $10^7$ steps of the engaged state. The translocation step size is estimated as the difference between the average base positions of the DNA in contact with ScpA during the first disengaged state and the average base positions of the DNA segment in contact with ScpA, which was pumped from the SMC ring, in the last disengaged state.

A successful translocation trajectory was defined as one in which the DNA loop is maintained within the SMC ring during the pumping of DNA in the last disengaged state, and where there are discontinuous jumps observed in the DNA base positions contacting with ScpA (*Figure 5I, J*, *Figure 5—figure supplement 4*). We defined DNA slippage as a trajectory in which DNA continues to contact with ScpA while sliding around the hinge, coiled-coil, and ATPase head domains, resulting in the gradual destabilization and eventual release of the DNA loop within the SMC ring.

### Analysis of spatial distribution of the DNA patch of the ScpA subunit

The spatial distribution of the ScpA subunit (*Figure 7C*, *Figure 7—figure supplement 1*) was analyzed following structural alignment of the SMC ATPase dimer, ensuring that the ring structure formed by the coiled-coil arms was parallel to the *xz*-plane. To eliminate biases from initial structural modeling, one complete ATPase cycle was executed, and the analysis was conducted on the engaged state during the second ATPase cycle.

## Acknowledgements

In this research work, we used the supercomputer of Academic Center for Computing and Media Studies (Kyoto University). This study was supported by JSPS KAKENHI, Grant Number 20H05934 (ST) and 21H02441 (ST) and by the MEXT grant JPMXP1020230119 as 'Program for Promoting Researches on the Supercomputer Fugaku' (ST).

## Additional information

### Funding

| Funder | Grant reference number | Author |
| --- | --- | --- |
| JSPS KAKENHI | 20H05934 | Shoji Takada |
| JSPS KAKENHI | 21H02441 | Shoji Takada |
| MEXT grants "Program for Promoting Researches on the Supercomputer Fugaku" | JPMXP1020230119 | Shoji Takada |

The funders had no role in study design, data collection, and interpretation, or the decision to submit the work for publication.

### Author contributions

Masataka Yamauchi, Conceptualization, Software, Validation, Investigation, Visualization, Methodology, Writing - original draft, Writing – review and editing; Giovanni Bruno Brandani, Tsuyoshi Terakawa, Investigation, Writing – review and editing; Shoji Takada, Conceptualization, Supervision, Investigation, Project administration, Writing – review and editing

## Author ORCIDs

Masataka Yamauchi ⓘ https://orcid.org/0000-0002-5123-3993
Giovanni Bruno Brandani ⓘ https://orcid.org/0000-0003-3379-0187
Tsuyoshi Terakawa ⓘ https://orcid.org/0000-0002-0151-1123
Shoji Takada ⓘ https://orcid.org/0000-0001-5385-7217

Reviewer #1 (Public review): https://doi.org/10.7554/eLife.106752.3.sa1
Reviewer #2 (Public review): https://doi.org/10.7554/eLife.106752.3.sa2
Author response https://doi.org/10.7554/eLife.106752.3.sa3

## Additional files

### Supplementary files

MDAR checklist

### Data availability

The primary inputs for the molecular dynamics simulations and representative trajectories generated during this study are publicly available on Zenodo at https://doi.org/10.5281/zenodo.19681714.

The following dataset was generated:

| Author(s) | Year | Dataset title | Dataset URL | Database and Identifier |
|---|---|---|---|---|
| Yamauchi M | 2026 | SMC complex unidirectionally translocates DNA by coupling segment capture with an asymmetric kleisin path | https://doi.org/10.5281/zenodo.19681714 | Zenodo, 10.5281/zenodo.19681714 |

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

## Appendix 1

## Validation of the reliability of all-atom force fields in the bottom-up coarse-graining approach

In this study, we adopted a bottom-up approach to model hydrogen-bond-like interactions between protein and DNA interactions in a coarse-grained model. In this approach, the accuracy of all-atom simulations potentially influences the description of the coarse-grained model. In particular, the hydrogen bond patterns might vary depending on the choice of force field set used in the all-atom simulations.

To validate the robustness of the force field parameter set used in this study (Amber ff19SB for proteins, *Tian et al., 2020*; OL15 for DNA, *Zgarbová et al., 2015*; and OPC for water, *Izadi et al., 2014*), we investigated force field dependency of the hydrogen bonding patterns between SMC ATPase heads and DNA.

In addition to the original parameter set (Amber ff19SB for proteins, OL15 for DNA, and OPC for water), we performed all-atom simulations in which the protein force field was changed to Amber ff99SB-ILDN (*Lindorff-Larsen et al., 2010*), the DNA force field to Bsc1 (*Ivani et al., 2016*), and the water model to TIP4P-D (*Piana et al., 2015*). Here, we adopted $Mg^{2+}$ ion parameters optimized for the TIP4P-D water model (*Grotz and Schwierz, 2022*). The result of the fraction of hydrogen-bond forming residues is shown in *Figure 1—figure supplement 2B*. Although there were variations in the fractions and rankings, the top six amino acid residues that form hydrogen bonds with DNA are common between the original and the second force field parameter sets: 120Arg, 123Arg, 63Ile, 111Arg, 62Arg, and 56Lys. Importantly, biochemical experiments have demonstrated the crucial roles of Arg62, Arg120, and Arg123 in the interaction between SMC ATPase heads and DNA (*Vazquez Nunez et al., 2019*). Even in amino acid residues with a lower fraction of hydrogen bond formation, we found common amino acid residues between two force field parameter sets such as 115Trp, 107Tyr, 105Arg, 124Ser, 54Ser, 119Arg, and 112Ser and so on. Such consistency between two force field parameter sets supports the reliability of the original force field set (Amber ff19SB + OL15 + OPC).

Furthermore, we conducted all-atom simulations with a third set of force fields: Amber ff99SB-ILDN for proteins (*Lindorff-Larsen et al., 2010*), bsc1 for DNA (*Ivani et al., 2016*), and TIP3P for water (*Jorgensen et al., 1983*). As shown in *Figure 1—figure supplement 2C*, we find that identified hydrogen-bond forming residues are essentially the same as in the original force field parameter set (Amber ff19SB + OL15 + OPC) and the second force field sets (Amber ff99SB-ILDN + Bsc1 + TIP4P-D).

These results clearly show that the choice of force field, water models, and magnesium ion parameters has only a minor impact on the hydrogen bond formation pattern between SMC ATPase and the DNA. Therefore, we conclude that the set of force fields we used in the bottom-up approach (i.e., Amber ff19SB + OL15 +OPC water) is reliable for capturing the key amino acid residues and modeling coarse-grained hydrogen bond-like interactions.

