## [Editor Report · eLife Assessment]

This **important** study presents a well-constructed multiscale simulation framework to investigate ATP-driven DNA translocation by prokaryotic SMC complexes, supporting a segment-capture mechanism. The strength of evidence is **convincing**, highlighting the necessity of a precise balance between electrostatic interactions and hydrogen bonding, as well as the critical role of kleisin asymmetry in ensuring unidirectional movement.

---

## [Referee Report · Reviewer #1 (Public review)]

Summary:

This study used explicit-solvent simulations and coarse-grained models to identify the mechanistic features that allow for unidirectional motion of SMC on DNA. Shorter explicit-solvent models provides a description of relevant hydrogen bond energetics, which was then encoded in a coarse-grained structure-based model. In the structure-based model, the authors mimic chemical reactions as signaling changes in the energy landscape of the assembly. By cycling through the chemical cycle repeatedly, the authors show how these time-dependent energetic shifts naturally lead SMC to undergo translocation steps along DNA that are on a length scale that has been identified.

Strengths:

Simulating large-scale conformational changes in complex assemblies is extremely challenging. This study utilizes highly-detailed models to parameterize a coarse-grained model, thereby allowing the simulations to connect the dynamics of precise atomistic-level interactions with a large-scale conformational rearrangement. This study serves as an excellent example for this overall methodology, where future studies may further extend this approach to investigated any number of complex molecular assemblies.

Comments on revisions:

No additional recommendations. I removed the weakness description in the summary, since the authors have addressed that concern.

---

## [Referee Report · Reviewer #2 (Public review)]

Summary:

The authors perform coarse grained and all atom simulations to provide a mechanism for loop extrusion that is involved in genome compaction.

Strengths:

The simulations are very thoughtful. They provide insights into the the translocation process, which is only one of the mechanisms. Much of the analyses is very good. Over all the study advances the use of simulations in this complicated systems.

Weaknesses:

Even the authors point out several limitations, which cannot be easily overcome in paper because of the paucity of experimental data. Nevertheless, the authors could have done to illustrate the main assertion that loop extrusion occurs by the motor translocating on DNA. They should mention more clearly that there are alternate theory that have accounted for a number of experimental data.

Comments on revisions:

The authors have adequately addressed my concerns.

---

## [Author Response]

The following is the authors’ response to the original reviews.

**Public Reviews:**

**Reviewer #1 (Public review):**
Summary:This study used explicit-solvent simulations and coarse-grained models to identify the mechanistic features that allow for the unidirectional motion of SMC on DNA. Shorter explicit-solvent models describe relevant hydrogen bond energetics, which were then encoded in a coarse-grained structure-based model. In the structure-based model, the authors mimic chemical reactions as signaling changes in the energy landscape of the assembly. By cycling through the chemical cycle repeatedly, the authors show how these time-dependent energetic shifts naturally lead SMC to undergo translocation steps along DNA that are on a length scale that has been identified.Strengths:Simulating large-scale conformational changes in complex assemblies is extremely challenging. This study utilizes highly-detailed models to parameterize a coarse-grained model, thereby allowing the simulations to connect the dynamics of precise atomistic-level interactions with a large-scale conformational rearrangement. This study serves as an excellent example for this overall methodology, where future studies may further extend this approach to investigated any number of complex molecular assemblies.

We thank the reviewer for careful reading of our manuscript and highlighting the value of our bottom-up multiscale simulation approach.

Weaknesses:The only relative weakness is that the text does not always clearly communicate which aspects of the dynamics are expected to be robust. That is, which aspects of the dynamics/energetics are less precisely described by this model? Where are the limits of the models, and why should the results be considered within the range of applicability of the models?

We appreciate this insightful comment and agree that it is important to more explicitly describe the robustness and limitations of the simulation model used in this study. In response to this comment, we have revised the Discussion section of our manuscript.

First, to clarify the robust aspects of our model, we have added a new subsection titled “Parametric choices and robustness of simulation model” to the Discussion, which is as follows:

“The switching Gō approach adopted in this study is a powerful tool for providing the relationship between known large-scale conformational changes and the resulting functional and mechanical dynamics of the molecular machine (Brandani and Takada, 2018b; Koga and Takada, 2006b; Nagae et al., 2025). In this study, we mimic conformational change induced by ATP binding and hydrolysis events by instantaneously switching the potential energy function from one that stabilized a given conformation to another that stabilized a different conformation. This drives the protein to undergo a conformational transition toward the minimum of the new energy landscape.

This approach is particularly well suited to investigate whether a given conformational change in a subunit of a molecular machine can produce the overall motion observed, and whether this process is mechanically feasible. Therefore, the fundamental mechanisms identified in this study, i.e., DNA segment capture mechanism, the correlation between step size and loop length, and the unidirectional translocation mechanism originating from the asymmetric kleisin path, can be considered as robust, as they emerge directly from the structural and topological constraints of the SMC-kleisin architecture rather than from tuned parameters.”

Additionally, to more clearly define the limits of our model, we have expanded the "Limitations in current simulations" subsection. Specifically, we have added a detailed discussion regarding the energetics and transition pathways inherent to the switching Gō approach, which is as follows:

“First, use of switching potentials to trigger conformational changes impose a limitation on predictive power for energetics and transition pathways. The switching of potentials is akin to a “vertical excitation” from one energy landscape to another, rather than a thermally activated crossing of an energy barrier. Consequently, the model cannot provide quantitative predictions of the transition rates or the free energy barriers associated with these changes. Furthermore, while the subsequent relaxation follows the new potential landscape, it is not guaranteed to reproduce the unique, physically correct transition pathway. Nevertheless, this simplification is justified because conformational changes within the protein are expected to occur on a much faster timescale than the large-scale motion of the DNA. Thus, this simplification has a limited impact on our main conclusions regarding the functional DNA dynamics driven by these large-scale conformational changes.”

We have not made any additions regarding the timescale and dwell times for each ATP state, as these were already discussed in the original manuscript.

**Reviewer #2 (Public review):**
Summary:The authors perform coarse grained and all atom simulations to provide a mechanism for loop extrusion that is involved in genome compaction.Strengths:The simulations are very thoughtful. They provide insights into the translocation process, which is only one of the mechanisms. Much of the analyses is very good. Over all the study advances the use of simulations in this complicated systems.

We sincerely thank the reviewer for their thoughtful and encouraging comments.

Weaknesses:Even the authors point out several limitations, which cannot be easily overcome in the paper because of the paucity of experimental data. Nevertheless, the authors could have done so to illustrate the main assertion that loop extrusion occurs by the motor translocating on DNA. They should mention more clearly that there are alternative theories that have accounted for a number of experimental data.

We thank the reviewer for these constructive suggestions. As the reviewer pointed out, it is important to state more explicitly how the unidirectional DNA translocation revealed in this study relates to the widely recognized loop-extrusion hypothesis of genome organization and situate our findings with the context of major alternative theories.

To address this, we first clarify the relationship between the translocation mechanism we observed and the phenomenon of loop extrusion. We emphasize that our simulations were designed to elucidate the core motor activity of the SMC complex, and we explicitly state our view that loop extrusion is a functional consequence of this motor activity when the complex is anchored to DNA.

Second, as the reviewer also suggested, we addressed alternative models of loop extrusion that also have experimental support in more details. We have revised the Discussion accordingly to provide a more balanced and comprehensive context. Further details are provided in our separate response to the comment below.

**Reviewer #3 (Public review):**
Summary:In this manuscript, Yamauchi and colleagues combine all-atom and coarse-grained MD simulations to investigate the mechanism of DNA translocation by prokaryotic SMC complexes. Their multiscale approach is well-justified and supports a segment-capture model in which ATP-dependent conformational changes lead to the unidirectional translocation of DNA. A key insight from the study is that asymmetry in the kleisin path enforces directionality. The work introduces an innovative computational framework that captures key features of SMC motor action, including DNA binding, conformational switching, and translocation.This work is well executed and timely, and the methodology offers a promising route for probing other large molecular machines where ATP activity is essential.Strengths:This manuscript introduces an innovative yet simple method that merges all-atom and coarse-grained, purely equilibrium, MD simulations to investigate DNA translocation by SMC complexes, which is triggered by activated ATP processes. Investigating the impact of ATP on large molecular motors like SMC complexes is extremely challenging, as ATP catalyses a series of chemical reactions that take and keep the system out of equilibrium. The authors simulate the ATP cycle by cycling through distinct equilibrium simulations where the force field changes according to whether the system is assumed to be in the disengaged, engaged, and V-shaped states; this is very clever as it avoids attempting to model the non-equilibrium process of ATP hydrolysis explicitly. This equilibrium switching approach is shown to be an effective way to probe the mechanistic consequences of ATP binding and hydrolysis in the SMC complex system.The simulations reveal several important features of the translocation mechanism. These include identifying that a DNA segment of ~200 bp is captured in the engaged state and pumped forward via coordinated conformational transitions, yielding a translocation step size in good agreement with experimental estimates. Hydrogen bonding between DNA and the top of the ATPase heads is shown to be critical for segment capturtrans, as without it, translocation is shown to fail. Finally, asymmetry in the kleisin subunit path is shown to be responsible for unidirectionally.This work highlights how molecular simulations are an excellent complement to experiments, as they can exploit experimental findings to provide high-resolution mechanistic views currently inaccessible to experiments. The findings of these simulations are plausible and expand our understanding of how ATP hydrolysis induces directional motion of the SMC complex.

We thank the reviewer for the thoughtful and encouraging assessment of our work. We appreciate the reviewer’s summary of our key contributions, especially our switching Gō strategy, the segment-capture mechanism of SMC translocation, and the role of kleisin-path asymmetry in ensuring unidirectionality.

Weaknesses:There are aspects of the methodology and modelling assumptions that are not clear and could be better justified. The major ones are listed below:(1) The all-atom MD simulations involve a 47-bp DNA duplex interacting with the ATPase heads, from which key residues involved in hydrogen bonding are identified. However, DNA mechanics-including flexibility and hydrogen bond formation-are known to be sequence-dependent. The manuscript uses a single arbitrary sequence but does not discuss potential biases. Could the authors comment on how sequence variability might affect binding geometry or the number of hydrogen bonds observed?

We thank the reviewer for this insightful comment regarding the potential effects of DNA sequence.

The primary biological role of the SMC complex is to organize genome architecture on a global scale; as such, its fundamental interaction with DNA is considered not to be sequence-specific. Our all-atom MD simulations and analysis pipeline were designed to probe the nature of this general interaction. Our approach confirms this rationale: the analysis exclusively identified hydrogen bonds formed between amino acid residues and the phosphate groups of the DNA's sugar-phosphate backbone. As shown in Figs. 1B and 1C, the results confirm that the key stabilizing interactions occur between basic residues on the SMC head surface and the DNA backbone. Since the backbone is chemically uniform, the stable binding mode we characterized is inherently sequence-independent.

While the final bound state is likely sequence-independent, we agree that sequence-dependent properties such as local DNA flexibility or intrinsic curvature could influence the kinetics of the binding process. For example, the rate of initial recognition or the ease of DNA bending on the head surface might vary between AT-rich and GC-rich regions. However, once the DNA is bound, we expect the stable binding geometry and the identity of the key interacting residues to be conserved across different sequences.

Therefore, we are confident that using a single, representative DNA sequence is a valid approach for elucidating the fundamental, non-sequence-specific aspects of SMC-DNA interaction and does not alter the general validity of the translocation mechanism proposed in this work.

(2) A key feature of the coarse-grained model is the inclusion of a specific hydrogen-bonding potential between DNA and residues on the ATPase heads. The authors select the top 15 hydrogen-bond-forming residues from the all-atom simulations (with contact probability > 0.05), but the rationale for this cutoff is not explained. Also, the strength of hydrogen bonds in coarse-grained models can be sensitive to context. How did the authors calibrate the strength of this interaction relative to electrostatics, and did they test its robustness (e.g., by varying epsilon or residue set)? Could this interaction be too strong or too weak under certain ionic conditions? What happens when salt is changed?

Thank you for these comments. We provide our rationale for the parameter choices below.

The contact probability cutoff of 0.05 was chosen to create a comprehensive set of residues that form physically robust interactions with DNA. To establish this robustness, we performed a parallel set of all-atom simulations using a different force field (see Fig. S2). This cross-validation revealed two key points. First, the top six residues (Arg120, Arg123, Ile63, Arg111, Arg62, and Lys56), which include experimentally confirmed DNA-binding sites, consistently exhibited the highest contact probabilities in both force fields, confirming the reliability of our identification. Second, and just as importantly, many residues with lower contact probabilities (e.g., Trp115, Tyr107, Arg105, Ser124, and Ser54) were also consistently detected across both simulations. This reproducibility suggests that these interactions are physically robust and not artifacts of a specific force field. We therefore concluded that a 0.05 cutoff is a well-balanced threshold that ensures the inclusion of not only the primary anchor residues but also the secondary, moderately interacting residues that are crucial for cooperatively stabilizing the DNA. We discussed this point in Method in the revised manuscript, which is as follows:

“The rationale for this cutoff is the physical robustness of the identified interactions; all-atom simulations using a different force field confirmed that the same set of key interacting residues, including both strong and moderate binders, was consistently identified (Fig. S2).”

The strength of the hydrogen bond potential was set to ϵ = 4.0 kT (≈2.4 kcal/mol), a physically plausible value corresponding to an ideal hydrogen bond. To test the robustness of this parameterization, we performed preliminary simulations where we varied these parameters by (i) reducing the value of ϵ and (ii) restricting the interaction to only the top six anchor residues. In both test cases, while a short DNA duplex (47 bp) could still bind to the ATPase heads, simulations with a long DNA (800 bp) failed to form a stable DNA loop after initial docking. These tests demonstrated that a larger set of cooperative interactions with a physically realistic strength was necessary for the full segment capture mechanism. Our final parameter set (15 residues at ϵ = 4.0 kT) was thus chosen as the parameter set required to capture both the initial anchoring of DNA and the subsequent cooperative stabilization of the captured loop.

As correctly pointed out, ionic conditions are a critical factor. Our simulations revealed that the salt concentration had a more pronounced effect on the kinetics of the DNA finding its correct binding site rather than on the thermodynamic stability of the final bound state. During our parameter tuning, we found that at physiological salt conditions (150 mM), long-range electrostatic interactions become dominant. This caused the DNA to be non-specifically captured by positively charged patches on the sides of the heads, which are not the functional binding sites. This off-pathway trapping kinetically prevented the DNA from reaching its proper location within the simulation timeframe. In contrast, the high-salt conditions (300 mM) used in this study screen these long-range interactions, suppressing non-specific trapping and allowing the DNA to efficiently explore the protein surface. This enables the correct binding to be established via the specific, short-range hydrogen bonds. Therefore, the ion concentration in our model is more as a crucial kinetic control factor to reproduce correct binding pathway within a realistic simulation timeframe. This point is discussed in the new subsection entitled “Parametric choices and robustness of simulation model”.

(3) To enhance sampling, the translocation simulations are run at 300 mM monovalent salt. While this is argued to be physiological for Pyrococcus yayanosii, such a concentration also significantly screens electrostatics, possibly altering the interaction landscape between DNA and protein or among protein domains. This may significantly impact the results of the simulations. Why did the authors not use enhanced sampling methods to sample rare events instead of relying on a high-salt regime to accelerate dynamics?

We agree that enhanced sampling methods are powerful for exploring rare events. However, many of these techniques require the pre-definition of a suitable, low-dimensional reaction coordinate (RC) to guide the simulation. The primary goal of our study was to discover the DNA translocation mechanism as it emerges naturally from fundamental physical interactions, without imposing a priori assumptions about the specific pathway.

The DNA segment capture process is complex, involving the coordinated motion of a long DNA polymer and multiple protein domains. Defining a simple RC in advance was not feasible and would have carried a significant risk of biasing the system toward an artificial pathway. Therefore, to avoid such bias, we chose to perform direct, unbiased molecular dynamics simulations. Using a physiologically relevant high-salt concentration (300 mM) for Pyrococcus yayanosii was a strategy to accelerate the system's natural dynamics, allowing us to observe these unbiased trajectories within a feasible computational timescale.

Because our current work has elucidated the fundamental steps of this mechanism, we agree that this work provides a foundation for more quantitative analyses. As suggested, future studies using methods like Markov State Model analysis or enhanced sampling techniques, guided by more sophisticated RCs defined from the insights of this work, would be a valuable next step for characterizing the free-energy landscape of the process or longer time scale dynamics.

(4) Only a small fraction of the simulated trajectories complete successful translocation (e.g., 45 of 770 in one set), and this is attributed to insufficient simulation time. While the authors are transparent about this, it raises questions about the reliability of inferred success rates and about possible artefacts (e.g., DNA trapping in coiled-coil arms). Could the authors explore or at least discuss whether alternative sampling strategies (e.g., Markov State Models, transition path sampling) might address this limitation more systematically?

We thank the reviewer for raising this point that is crucial for considering limitations and future directions of our study.

As we noted in a previous response, the primary reason we did not employ such enhanced sampling methods was the limited prior knowledge available to define previously uncharacterized DNA translocation process. Therefore, we first try to define the key conformational states and transitions without the potential bias of a predefined model or reaction coordinate. This approach was successful, as it allowed us to identify critical on-pathway states like “DNA segment capture” and significant off-pathway or kinetically trapped states such as 'DNA trapping' between the coiled-coil arms.

We fully agree that the low success rate observed is a key finding that points to significant kinetic bottlenecks, and that a more systematic analysis is required. Having identified the essential states, applying techniques such as Markov State Models (MSMs) or transition path sampling represents a powerful and logical next step. These methods, using a state-space definition based on our findings, will enable a quantitative characterization of the free-energy landscape and the transition rates between states. This will provide a rigorous understanding of the kinetic factors, such as the depth of the trapped-state energy well, that underlie the low translocation efficiency.

In the revised manuscript, we discuss the application of these advanced sampling methods as a feasible and promising future direction, which is as follows:

“Future studies can leverage the insights from this work to overcome the current timescale limitations. Techniques such as Markov state modeling (Husic and Pande, 2018; Prinz et al., 2011) or enhanced sampling methods (Hénin et al., 2022) may be employed to quantitatively characterize the free-energy landscape and transition rates. Such an approach would provide a rigorous understanding of the kinetic barriers, such as the stability of the trapped state, that govern the efficiency of SMC translocation.”

**Reviewer #1 (Recommendations for the authors):**
As noted in the public review, there could be a more systematic description of the limits of the model. The model appears to be carefully crafted, though every model has limits. It could be helpful for the general readership to give some idea of which parametric choices are more critical, and which mechanistic features should be robust to minor changes in parameters.

We sincerely thank the reviewer for this constructive comment. We agree that clarifying which aspects of our model is robust and sensitive to specific parameter choices is crucial for the reader's understanding.

We have expanded the Discussion to clarify how specific simulation parameters affect the efficiency and success rate of DNA translocation in our coarse-grained simulations. In particular, we have added a description of the parametric choices for (i) selection and strength of hydrogen bonds, (ii) ionic strength, and (iii) interaction strength between the coiled-coil arms. The discussion can be found in subsection entitled “Parametric choice and robustness of simulation model” in the Discussion, which is as follows:

“On the other hand, the efficiency and success rate of DNA translocation in our simulations are more sensitive to certain parametric choices. For instance, the selection and strength of hydrogen bond-like interactions are a key factor. Our model incorporates specific hydrogen bonds between the upper surface of the ATPase heads and DNA, based on all-atom simulations. These interactions are essential for initiating segment capture; without them, DNA fails to migrate to the correct binding surface. While the identification of these key residues is a robust finding—persisting across different all-atom force fields (Fig. S2)—their strength and number in the coarse-grained potential are critical parameters that directly influence the probability and kinetics of DNA capture. Another critical parameter is the ionic strength. We performed translocation simulations at an ionic strength of 300 mM to accelerate DNA dynamics. At lower concentrations, non-specific electrostatic interactions between DNA and positively charged patches on the sides of the ATPase heads or coiled-coil arm became dominant, hindering the efficient migration of DNA to its functional binding site. Using a higher-than-physiological ionic strength is a justified practice in coarse-grained simulations employing the Debye-Hückel approximation, as it serves as a first-order correction to mimic the strong local charge screening by condensed counterions that is not explicitly captured by the mean-field model (Brandani et al., 2021; Niina et al., 2017b). Finaly, the interaction strength between the coiled-coil arms is also important. In our model, once the arms closed during the transition from the V-shaped to the disengaged state, they remained closed on the simulated timescale, frequently trapping DNA pushed from the hinge and thereby leading to failed translocation. This behavior suggests that the arm–arm interactions may be overestimated. A parameterization that allows for more frequent, transient opening of the arms could increase the success rate of DNA pumping.”

**Reviewer #2 (Recommendations for the authors):**
This paper reports simulations (all atom and coarse grained) to provide molecular details of loop extrusion. In general, it is a well done paper. There are a few issues that the authors should address.(1) The study supposes that loop extrusion occurs by translocation. Although they point out alternate models like scrunching (C Dekker; the theory by Takaki is also based on the scrunching model that the authors should mention), they should discuss this further. After all, the Takaki theory does predict several experimental outcomes very accurately. The precise mechanism has not been nailed down - The paper by Terakawa in Science suggests the extrusion is by translocation, but the evidence is not clear.

We thank the reviewer for this insightful comment. We agree that our discussion should briefly acknowledge alternative models such as scrunching. We have therefore revised the manuscript to mention the theory by Takaki et al. (Nat. Commun., 2021), which reproduces several experimental outcomes.

Because our present work specifically addresses the translocation mechanism based on DNA segment capture, we now state that scrunching and related models represent alternative proposals for loop extrusion.

In this revision, we have added discussion to the end of the subsection titled "DNA segment capture as the mechanism of the DNA translocation by SMC complexes." in the Discussion section, which is as follows:

“Turning to loop extrusion mechanisms, alternative mechanisms have been proposed in addition to the DNA-segment capture model. For example, Takaki et al. developed a scrunching-based theory that quantitatively accounts for several experimental observations, including force-velocity relationships and step-size distributions. While our present study focuses on the DNA translocation mechanism via segment capture, it is important to note that scrunching and other models remain plausible alternatives for loop extrusion. The precise mechanism may depends on the specific SMC complex and their subunits and remains to be fully resolved.”

(2) It is unclear how one can say from Figure 4I and J that translocation has taken place. These panels show that the base pair length increases. This should be explained more clearly. They should also simultaneously plot the location of the heads (2D plot).

Thank you for this valuable suggestion. In response to the comment on how translocation is presented in Fig. 4I and J, we have revised the text to make it clear that the SMC complex moves along DNA in subsection entitled “DNA translocation via DNA-segment capture”, as follows:

“Fig. 4I represents the one-dimensional contour coordinate of the DNA molecule, indexed by base pairs (1-800). In this plot, translocation is visualized as a discontinuous shift in the range of base-pair indices that the SMC complex contacts over one complete ATP cycle”

“This translocation is recorded in Fig. 4I as the average coordinate of the kleisin contact region (red dots) jumps from ~400 bp before the cycle to ~600bp after, which corresponds to a translocation event of ~200 bp”

We believe that adding this explanation makes it clearer to readers that Fig. 4I and 4J provide direct evidence for unidirectional translocation of the SMC complex.

(3) The transitions between the states are very abrupt (see Figure 2). Please explain. Also, in which state does extrusion take place? What is the role of the V-shape - is it part of the ATPase cycle?

We thank the reviewer for raising these questions.

In our simulation, we implemented ATP-binding state change by instantaneously switching the structure-based (Gō-type) potential between reference conformations for the disengaged (apo), engaged (ATP-bound), and V-shaped (ADP-bound) states at predetermined times. The system rapidly relaxes along the new funnel-shaped potential energy surface toward its minimum. This rapid relaxation is why the transition appears abrupt in metrics such as the Q-score in Fig.2.

The V-shaped state corresponds to a key ADP-bound intermediate within the ATP hydrolysis cycle. Its primary role in our model is preparatory; it establishes the necessary open geometry that allows for the subsequent "zipping" of the coiled-coil arms. Crucially, unidirectional pumping motion is generated during the transition from the V-shaped state to the disengaged state. That is, the zipping motion of the coiled-coil arm pushes the captured DNA segment forward, resulting in a net translocation along the DNA.

(4) It appears the heads do not move between the disengaged to engaged states. Why not in their model?

Thank you for pointing out the lack of clarity in explanation of the SMC head movement in our simulations.

In our model, the transition from the disengaged to the engaged state involves a dynamic rearrangement of the SMC heads. Specifically, one ATPase head slides (~10 Å) and rotates (~85°) relative to the other ATPase head to re-associate at a new dimer interface. This movement drives the global conformational change of the complex from a rod-like shape to an open ring, a mechanism proposed in a previous structural study (Diebold-Durand et al., Mol. Cell, 2017).

As reviewer 2 noted, this crucial motion, which is reflected in the changing head-head distance and hinge angle in Fig. 2A, was not sufficiently highlighted in the text. We have therefore revised the manuscript to explicitly describe this head rearrangement to improve clarity, which is as follows:

“Upon transition to the engaged state, the two ATPase heads were quickly rearranged to form the new inter-subunit contacts. Specifically, this rearrangement involves one ATPase head sliding by approximately 10 Å and rotating by 85° relative to the other, allowing it to associate through a different interface (Diebold-Durand et al., 2017b). The fractions of formed contacts, Q-scores, that exist at the disengaged (engaged) states quickly decreased (increased) (Fig. 2A, top two plots).”

(5) What is pumping - it has been used in Marko NAR in the DNA capture model. How is that illustrated in the simulations?

We thank the reviewer for raising this point. In the context of the DNA segment-capture model by Marko et al. (NAR, 2019), "pumping" refers to the conceptual process where a DNA loop, captured in an upper compartment of the SMC ring, is transferred to a lower compartment, resulting in net translocation.

Our simulations provide a direct, molecular-resolution visualization of the physical mechanism underlying this concept. We illustrate that the "pumping" action is not a passive transfer but an active, mechanical process driven by a specific conformational change. This occurs during the transition from the V-shaped (ADP-bound) to the disengaged state. As shown in our trajectories, the two coiled-coil arms close in a zipper-like manner, beginning from the hinge and progressing toward the ATPase heads. This zipping motion physically pushes the captured DNA segment from the hinge region toward the kleisin ring.

This process is visualized in our simulations as a clear, unidirectional translocation step (see Figs. 4B–D, 4I, and S6). The result is a net forward movement of the DNA by a distance that corresponds to the length of the initially captured loop, a key prediction of the Marko’s model that we quantify in our step-size analysis (Figs. 4K–L and S8).

To make this point clearer for the reader, we have revised the manuscript. We have explicitly defined this "zipping and pushing" action as the physical basis for the "pumping" mechanism in the subsection titled "Zipping motion of coiled-coil arms pushes the DNA from hinge domain toward kleisin ring", which is as follows:.

“This active, mechanical pushing of the DNA loop, driven by the sequential closing of the coiled-coil arm, constitutes the physical basis of the “pumping” mechanism that drives unidirectional translocation. Our simulations thus provide a concrete, molecular-level visualization for this key step in the DNA segment-capture model.”

(6) The length of DNA simulated is small for understandable reasons. Both experiments and theory show that loop extrusion sizes can be very large, far exceeding the sizes of the SMA complex. Could the small size of DNA be affecting the results?

We thank the reviewer for this important comment. The relationship between our simulated system size and the large-scale phenomena observed experimentally is a key point.

Our study was specifically designed to elucidate the fundamental mechanism of the elementary, single-cycle translocation step at near-atomic resolution. For this purpose, the 800 bp DNA length was sufficient. The observed translocation step size per cycle was 216 ± 71 bp, which is substantially smaller than the total length of the simulated DNA. This confirms that the boundaries of our system did not artificially constrain the core translocation process we aimed to investigate. Therefore, we think that the DNA length used in this study did not systematically bias our main findings regarding the motor mechanism itself.

As the reviewer pointed out, on the other hand, our current setup cannot reproduce the formation of kilobase-scale loops. We hypothesize that these large-scale events are intrinsically linked to the stochastic nature of the ATP hydrolysis cycle, which was simplified in our simulation model. We used fixed durations for each state for computational feasibility. In a more realistic scenario, a stochastically prolonged engaged state would provide a larger duration time for a captured DNA loop to grow via thermal diffusion. This could lead to occasional, much larger translocation steps upon ATP hydrolysis, contributing to the large loop sizes seen experimentally.

(7) Minor point: The first CG model using three sites was introduced in PNAS vol 102, 6789 2005. The authors should consider citing it.

Thank you for this suggestion. We have now cited the paper the reviewer recommended. Please find subsection entitled Coarse-grained simulations in Materials and Methods.